



# Influence of initial soil moisture in a Regional Climate Model study over West Africa: Part 1: Impact on the climate mean.

Brahima KONÉ[1], Arona DIEDHIOU[1, 2], Adama Diawara[1], Sandrine Anquetin[2], N'datchoh Evelyne Touré[1], Adama Bamba[1], and Arsene Toka Kobea[1]

[1]LAPAMF, Université Félix Houphouët Boigny, Abidjan, Côte d'Ivoire

[2]Univ. Grenoble Alpes, IRD, CNRS, Grenoble INP, IGE, F-38000 Grenoble, France

*Correspondence to:* Arona DIEDHIOU (arona.diedhiou@ird.fr)

**Abstract.**

The impact of the anomalies in initial soil moisture in later spring on the subsequent mean climate over West Africa is examined using the latest version of Regional Climate Model of the International Centre for Theoretical Physics (RegCM4). We performed this sensitivity studies over the West African domain, for June-July-August-September (JJAS) 2003 (wet year) and JJAS 2004 (a dry year) at the horizontal resolution of 25 km × 25 km. The reanalysis soil moisture of the European Centre Meteorological Weather Forecast's reanalysis of the 20th century (ERA20C) were used to initialize the control runs, whereas we initialized the soil moisture at the wilting points and field capacity respectively in dry and wet experiments. The impact of the anomalies in initial soil moisture on the precipitation in West Africa is homogeneous only over the central Sahel where dry (wet) experiments lead to rainfall decrease (increase). The strongest impact on precipitation in wet and dry experiments is found respectively over west and central Sahel with the peak of change about respectively 40% and -8%. The impact of the anomalies in initial soil moisture can persist for three or even four months, however the significance influence on precipitation, greater than 1mm.day-1, of the impact of the anomalies in initial soil moisture is much shorter, no longer than one month. The effect of soil moisture anomalies is mostly confined to the near-surface climate and in the upper troposphere. Overall, the impact of the anomalies in initial soil moisture is greater on temperature than on precipitation over most areas studied. The strongest homogeneous impacts of the anomalies in initial soil moisture on temperature is located over the central Sahel with the peak of change at -1.5 °C and 0.5°C respectively in wet and dry experiments. The influence of initial the anomalies in initial soil moisture on the precipitation mechanism is also



highlighted. We will investigate in the Part II of this study the influence of the anomalies in
initial soil moisture on climate extremes.

**1 Introduction**

In the climate system soil moisture is one of the crucial variables which influence water balance
and surface energy components through latent surface fluxes and evaporation. Therefore, soil
moisture impacts the development of weather patterns and precipitation production. The
strength of soil moisture impact on land-atmosphere coupling is variable according to the place
and with season. Koster et al. (2004) sustained that the atmospheric response simulation to the
slow variation of the ocean and land surface states, can accurate the seasonal simulations. The
atmosphere response to ocean temperature anomalies is well documented (Kirtman and al.
1998; Rasmusson and al.1982). Another earth system component, potentially useful, that varies
slowly is soil moisture. The role of the soils may be comparable to that of the oceans. While in
summer, the solar energy received by the oceans is stored (and use it to heat the atmosphere in
winter), in winter the precipitation received by the soil is stored (the moistening and cooling is
return to the atmosphere in summer). Through its impact on surface energy fluxes and
evaporation, there are many additional impacts on climate process of soil moisture, such as
boundary-layer stability and air temperature (Hong and al., 2000; Kim and Hong 2007). Several
studies shown that, the anomalies of the soil moisture may persist for several weeks or months,
however, its impact remains only for shorter time in the atmosphere, not exceed few days
(Vinnikov and Yeserkepova 1991; Liu and al., 2014). The important role of the anomalies in
soil moisture in coupling of land and atmosphere is shown in several studies, using numerical
climate models (Jaeger and al., 2011; Zhang and al., 2011) and observation datasets (Zhang et
al., 2008a; Dirmeyer et al., 2006).
West Africa known to be a region where there is a strong coupling between soil moisture and
precipitation (Koster et al., 2004). Several previous studies have been conducted over West
Africa on a global scale using AGCMs (Atmospheric General Circulation Model) to investigate
the impact on land-atmosphere coupling of soil moisture anomalies (Koster and al., 2004;
Douville and al, 2001; Zhang and al., 2008b). However at the local and regional scales, the
land-atmosphere coupling studies with AGCM, present large uncertainties (Xue et al. 2010).
Recently, the use of RCMs to simulate the impact on interannual climate variability of
anomalies in soil moisture received a lot of attention because of the increase in climate
variability associated with extreme weather events that can have greater societal and
environmental impacts. In general, these studies have been conducted for Asia, Europe and



America (e.g. Seneviratne and al. 2006 for Europe; Zhang and al. 2011 for Asia; Zhang and al.
2008b for America). Overall, the results of these studies show, during summertime, the strong
impact of the anomalies of soil moisture in land-atmosphere occurred mainly over the transition
zones with a climate between wet and dry climate regimes. The relevance and extent of this
potential feedback are still poorly understood over the West Africa. This study will focus on
the influence of initial soil moisture anomalies on climate mean and it is based on performance
assessment of the Regional Climate model version 4 coupled to the version 4.5 of the
Community Land Model (RegCM4-CLM4.5) done by Koné and al. (2018) where the ability of
the model to reproduce the climate mean has been validated. While in the part II of the article,
the influence of soil moisture on climate extremes will be explored. The descriptions of the
model and experiment setup used in this study are presented in Section. 2; in the Section 3, the
influence of the anomalies in initial soil moisture on the subsequent climate mean is analyzed
and discussed; and in Section 4 the main conclusions close the paper.
**2. Model and experimental design**
**2.1 Descriptions of the model and the observed datasets**
We used in this study, the fourth generation of the Regional Climate Model (RegCM4) of the
International Centre for Theoretical Physics (ICTP). Since this release, the physical
representations have been submitted to a continuous process of development and
implementation. The version used in the present study is RegCM4.7. The MM5 (Grell et al.,
1994) non-hydrostatic dynamical core has been ported to RegCM without removing the existing
hydrostatic core. The model dynamical core used in this study is the non-hydrostatic. RegCM4
is a limited area model using a sigma pressure vertical grid and the finite differencing algorithm
of Arakawa B-grid (Giorgi and al., 2012). The radiation scheme used in this version of
RegCM4.7 is derived from NCAR (National Center for Atmospheric Research) Community
Climate Model Version 3 (CCM3) (Kiehl and al., 1996), the representation of aerosols is from
Zakey and al. (2006) and Solmon and al. (2006). The scheme of the large-scale precipitation
used is from Pal and al. (2000), the moisture scheme is the SUBEX (SUBgrid EXplicit moisture
scheme) takes in account the cloud variability scale sub-grid, and the accretion processes and
evaporation for stable precipitation following the work of Sundqvist and al., 1989. In planetary
boundary layer, the sensible heat over ocean and land, the water vapor and the turbulent
transports of momentum are calculated according to the scheme of Holtslag and al. (1990). The
heat and moisture, the momentum fluxes of ocean surfaces in this study are computed as in
Zeng and al. (1998). In RegCM4.7, convective precipitation and the land surface processes can





be described by several parameterizations. Based on Koné and al. (2018), we selected the
convective scheme of Emanuel (Emanuel, 1991) and the interaction processes between soil,
vegetation and atmosphere are parameterized with CLM4.5. In each grid cell, CLM4.5 has 16
different PFTs (Plant functional Types) and 10 soil layers (Lawrence et al., 2011; Wang and
al., 2016). RegCM4 was integrated over the domain of West Africa depicted in Fig. 1 with 25
km of horizontal resolution and with 18 vertical levels and the initial and boundary conditions
are from the European Centre for Medium-Range Weather Forecasts reanalysis (EIN75; Uppala
and al., 2008; Simmons and al., 2007). The Sea Surface Temperatures (SST) is from the
National Oceanic and Atmosphere Administration (NOAA) optimal interpolation weekly
(OI_WK) (Reynolds and al., 1996). The source for the topography is from States Geological
Survey (USGS) Global Multi-resolution Terrain Elevation Data (GMTED; Danielson and al.,
2011) at 30 arc-second spatial resolution which is an update to the Global Land Cover
Characterization (GTOPO; Loveland and al., 2000) dataset.
Our analysis is focused on the precipitation and the 2m air temperature over the West African
domain during the summer of June-July-August-September (JJAS) for 2003 and 2004. The
uncertainties reduction related to the absence of reliable observation system over the region
(Sylla et al., 2013a; Nikulin et al.,2012), we validated the simulated precipitation based on two
products : the TRMM datasets (Tropical Rainfall Measuring Mission 3B43V7) at the high-
resolution 0.25°, available from 1998 to 2013 (Huffman and al., 2007), and The Climate
Hazards group Infrared Precipitation with Stations (CHIRPS) dataset developed at the
University of California at Santa Barbara at the 0.05° high-resolution available from 1981 to
2020. The validation of the simulated 2 m temperature relies on two observational datasets: the
global daily temperature from the Global Telecommunication System (hereafter GTS), gridded
at 0.5° of horizontal resolution for 1979 to 2020 (Fan and van den Dool, 2008) and the CRU
datasets (Climate Research Unit version 3.20) from the University of East Anglia, gridded at
the horizontal resolution 0.5°and available from 1901 to 2011 (Harris et al., 2013). To facilitate
the comparison with RegCM4 simulations, all products are re-gridded to 0.22° × 0.22° using a
method of bilinear interpolation (Nikulin et al., 2012).

### 2.2 Experiments setup and analysis methodology

It is known that sensitivity of initial soil moisture is no longer than one season. That's why
Hong and Pan (2000) use in their study only two years (3 months per year) to investigate the
impact of initial soil moisture over the North of America (in the Great Plains) during the two



summers, May-June-July (MJJ) 1988 (corresponding to a drought season) and MJJ 1993
(correspond to an extreme wet season). Over Asia also, Kim and Hong (2006) used in their
study two contrasted years (1997 and 1998, 4 months per year). In this study, the two years
2003 and 2004 have been chosen because they correspond respectively to a wet and dry year in
the region of interest. The simulations start from June $1^{st}$ and span four months, JJAS 2003 and
JJAS 2004, and the results during the first 7 days (Kang and al., 2014) are excluded in the
analysis as a spin-up period.
With very little variation in soil moisture in one day, initial soil moisture anomalies are given
at the first step of June $1^{st}$ for the two summers JJAS 2003 and JJAS 2004. Except for the
geographical location, the experimental setup is the same as that of Hong and al. (2000). The
geographical location of this study is the same as in Koné et al (2018), with four sub-regions,
each with different features of annual cycle of precipitation (Fig. 1). For each year, three
experiments are conducted; we used the soil moisture from the reanalysis of the European
Centre Meteorological Weather Forecast's reanalysis of the $20^{th}$ century (ERA20C) to initialize
the control runs. We initialized the dry and wet soil moisture (in volumetric fraction $m^3.m^{-3}$)
respectively at the wilting point (=$0.117*10^{-4}$) and the field capacity (=0.489) derived from
ERA20C dataset.
Generally, in several previous studies (Liu and al. (2014), Hong and al. (2000), Kim and Hong
(2006)), the analysis methodology used is the mean biases (MB) averaged over their domains
studied to quantify the impact of soil moisture anomalies, while in our study we used the mean
biases and the probability density function (PDF, Gao et al. 2016; Jaeger and Seneviratne 2011)
by fitting a normal distribution for this purpose to better capture how many grid points are
impacted by the anomalies in initial soil moisture. The pattern correlation coefficient (PCC) is
also used as spatial correlation to reveal the degree of large-scale similarity between model
simulations and the observation. We used the two-tailed t-test to investigate the differences
which are statistically significant at each grid cell between the control and the wet and dry
sensitivity experiments.

## 3. Results and discussion

### 3.1. Influence of initial soil moisture anomalies on precipitation.

Fig.2 displays the spatial distribution of observed mean rainfall (mm/day) from CHIRPS
(Fig.2a, d) and TRMM (Fig.2b, e) for JJAS 2003 and JJAS 2004  and their corresponding
simulated from control experiments (Fig.2c, f) initialized with reanalysis soil moisture


ERA20C. Table 1 reports the MB and the PCC for the simulations of the model and TRMM
observation compared to CHIRPS, computed for central Sahel, Guinea coast, west Sahel and
the entire West African domain. CHIRPS product displays a zonal band of rainfall centered
around 10° N decreasing from north to south (Fig.2a, d). The maximum values are located over
the mountain regions of Cameroun and Guinea. The TRMM observation (Fig.2b, e) is closer to
CHIRPS, and represents quite similarly the North–South gradient of precipitation with PCC up
to 0.97 over the entire West African domain for both JJAS 2003 and JJAS 2004 (Table 1).
However, although the observation datasets have similar large-scale patterns, they present
differences at the local scale. CHIRPS shows a much larger extend of these maxima than
TRMM, especially over the Guinea highland and Cameroon mountains, while TRMM shows a
large band of precipitation which extend too far into the Sahel region. The strongest mean bias
between the two products is dryer about -15.45 and -16.96 % respectively for JJAS 2003 and
JJAS 2004, and it is found over the Guinea coast sub-region (Table 1). The control experiments
(Fig.2 c and f) initialized with the reanalyze ERA20C soil moisture well reproduce the large-
scale pattern of the observed rainfall associated with PCC 0.72 and 0.77 (Table 1) respectively
for JJAS 2003 and JJAS 2004 over the West Africa domain, despite some biases at the locale
scale. The spatial extent of rainfall maxima and the North-South gradient are well captured by
control experiments; however, their magnitudes are underestimate.  In general, a dry mean bias
about -49.31% and -50.56% are found respectively for JJAS 2003 and JJAS 2004 over the
whole West African domain (Table 1). Figure 3 displays change in mean precipitation (in %)
for JJAS 2003 and JJAS 2004, for dry and wet experiments with respect to their corresponding
control experiments, the dotted area shows changes with statistical significance of 0.05 level.
The sensitivity dry and wet experiments show that precipitation has been significantly affected
by soil moisture anomalies at varying degrees according to the sub-regions (Fig. 3). In the dry
experiments (Fig.3a, c), we found a dominant decrease of rainfall over the central Sahel
especially in JJAS 2003 (Fig.3a), while the extent of this decrease is smaller in JJAS
2004(Fig.3c) and confined over the southern-west of Mali. On the other hand, we found a
dominant increase of rainfall over the Guinean coast and west Sahel, although there is a sparse
decrease, especially over the Guinea coast. In the wet experiments (Fig.3b, d), there is a
dominant increase of rainfall over most of the domains studied with a sparse decrease especially
along the coastline of Liberia, Sierra Leone and Guinea for both JJAS 2003 and JJAS 2004
(rep. Fig.3a and c). Overall, the impact on the precipitation of the anomalies in initial soil
moisture is homogeneous particularly over central Sahel, i.e, the dry (wet) experiments with
respect to the control exhibits significant decrease (increase) of precipitations (Fig.3a, b).





For a better quantitative evaluation, the PDF distributions of the changes in precipitation in
JJAS 2003 and JJAS 2004, over (a) central Sahel, (b) West Sahel, (c) Guinea coast and (d) West
Africa derived from dry and wet experiments compared to the corresponding control
experiments are shown in Figure 4. The impact on the precipitation of anomalies in initial soil
moisture is not homogeneous over most of the studied domains (Fig.4 b-d) except over central
Sahel where the dry (wet) experiments with respect to the control display significant decrease
(increase) of precipitation (Fig.4a,). However, the strongest impact on precipitation in wet and
dry experiments is found respectively over west and central Sahel, and the peak mode of change
is about respectively 40% and -8%. The impact on precipitation in wet experiment is stronger
than in dry experiment.
It is worth to note that, over the West Sahel and Guinea coast, for both dry and wet experiments
tend to cause an increase of precipitation. This indicates that the increase of precipitation is
more likely to happen not only in the wet experiment but also in the dry experiment (Fig. 4b).
The lag between the JJAS 2003 and JJAS 2004 PDFs for wet and dry experiments indicates a
somewhat significant impact when comparing the two years, particularly over Guinea and west
Sahel (Fig. 4 b and c). The wet year has a higher impact compared to the dry year over most of
domains studied (Fig. 4). These results are consistent with previous studies which supported a
strong relationship between precipitation and soil moisture in particular over the transition
zones with a climate between wet and dry climate regimes (Koster and al., 2004; Liu and al.,
2014; Douville and al., 2001).
To better study the influence of soil moisture anomalies on precipitation for the both dry and
wet years over the West African domain and its sub-regions, we analyzed changes in the daily
domain-average of soil moisture and precipitation (resp. Figure 5 and Figure 6) for JJAS 2003
and JJAS 2004, from dry and wet experiments with respect to their corresponding controls
experiments. The third soil layer in CLM4.5 (0 to 11.89 cm) is used in this study, this soil layer
corresponds almost to the top layer soil moisture used by Hong and al. (2000) in their work. In
general, soil moisture anomalies persist for three or four months over the domains studied
(Fig.5). The anomalies of soil moisture disappear for dry and wet experiments with varying
duration, between three to four months from one region to another over the domain studied.
The strongest duration and amplitude is found over the west Sahel sub-region, for the both wet
and dry experiments, it lasts four months in JJAS 2003 and JJAS 2004, although the signal is
rather weak in the wet experiments as compared to the dry ones (Fig. 5b). The weaker change
in soil moisture anomalies is found over the Guinea coast for wet experiments and lasts three



months (Fig. 5c). While in dry experiments, the weaker change in soil moisture anomalies is
found over central Sahel and last three months (Fig.5a).
Figure 6 shows response of the daily precipitation to the anomalies in initial soil moisture over
the different domains studied. In general, the impact of the wet soil moisture anomalies on daily
precipitation is larger in magnitude as compared to the dry anomalies over most of domains
studied (Fig. 6). The strongest daily precipitation response in dry experiment (-4mm.day$^{-1}$) is
found over the Guinea coast in the wet year JJAS 2003 (Fig. 6c), while for the wet experiments
(more than 8mm/day, especially in JJAS 2003), it is found over the West Sahel and the Guinea
coast (resp. Fig. 6 b and c). However, the impact on daily precipitation of the anomalies in
initial soil moisture is much shorter lived as compared to soil moisture change. The significant
impact on daily precipitation, greater than 1mm.day$^{-1}$ is shown only in wet experiment and last
no longer than fifteen days for most of domains studied, except for the Guinea coast where it
lasts about 1 month. It is worth to note the peaks in precipitation over West Sahel and Guinea
coast (resp. Fig.6b and c) during the two months August and September that coincide with
fluctuation in the anomalies of soil moisture (Fig.5b and c). This indicates the soil moisture and
precipitation feedback is strong during this period over Guinea coat and West Sahel regions.
The response of the daily precipitation to the anomalies in initial soil moisture is also sensitive
to the wet and dry year. This is indicated by the lag between dry and wet experiments for JJAS
2003 and JJAS 2004 years (Fig6). The magnitude of impacts due to contrasting years depends
on the place. For example, over Guinea coast, in the dry experiments, the wet year presents the
greater impact compared to the dry year (Fig.6 c). This trend is reversed for the central Sahel
(Fig. 6a). These results are in line with the previous works which argued that the soil moisture-
atmosphere feedback strength and the land memory depend on the place (Vinnikov et al. 1996;
Vinnikov and Yeserkepova 1991).
Figure 7 and 8 show the vertical profile change respectively in humidity and temperature for
JJAS 2003 and JJAS 2004 from the dry and wet experiments with respect to control experiments
over the whole West Africa domain and its sub-region indicated in Fig. 1.
For the dry and wet experiments, the impact on humidity and temperature (Fig. 7 and Fig. 8)
are significant in the lower troposphere. The dry (wet) soil moisture experiments, in the lower
and somewhat in the middle troposphere, show drying (moistening) and warming (cooling)
respectively for humidity and temperature, indicating weak (strong) dry convection over most
of the domains studied (Fig.7 and Fig.8). The strongest impact on humidity and temperature in
lower and middle troposphere is found over central Sahel (Fig. 7a and Fig. 8a). These results in
the lower troposphere are consistent with the precipitation sensitivity, especially over central





Sahel in JJAS 2003 (Fig.3 a, b). However, over west Sahel and the Guinea coast this impact is
somewhat weak as compared to central Sahel. In the dry experiments over the Guinea coast
(Fig. 7c), these trends are reversed above 500 hPa for humidity, indicating wet convection in
this sub-region.  These results in the lower atmosphere are consistent with the precipitation
sensitivity over the Guinea coast (Fig.3a, c).
On the over hand, in the upper troposphere, the significant impact on humidity and temperature
is found only for wet experiments, and exhibits a drying and warming respectively for humidity
and temperature over all the domains studied (Fig.7 and Fig.8). In the wet experiments, the
impact on upper tropospheric variability of the anomalies in initial soil moisture is also
identified by Hong and Pal (2000). The effect of soil moisture anomalies is mostly confined to
the near-surface and somewhat in the upper troposphere.
For understand of the origins of the precipitation changes in Figure 3, we analyzed the lower
tropospheric wind (850hpa) and moisture changes for JJAS 2003 and JJAS 2004 from the dry
and wet experiments with respect to their corresponding control experiments in Figure 9. In the
dry experiments, we found a dominant decrease of moistening over most of domain studied,
however the strong wind magnitude change over the Atlantic ocean, tends to bring the increase
of moistening from the ocean to Guinea coast and west Sahel, this can explain the increase  of
precipitation over these sub-region in the dry experiments. While over the central Sahel, a weak
change in wind magnitude is found, leading to the  strong decrease of precipitation there,
especially in JJAS 2003 (Fig.3a). On the other hand, in wet experiments, an increase in
moistening is located over most of domain studied. And the strong change in wind magnitude,
tend to bring the moistening from the North to South, leading to the increase of precipitation
observed over most of domain studied in wet experiments (Fig. 3 b and d). These results are
broadly consistent to the dry and wet precipitation changes shown in Figure 3.
Summarizing this section results, the anomalies in soil moisture anomalies persist for three or
four months, while the significant impact on precipitation, greater than 1mm.day$^{-1}$, of the
anomalies in soil moisture is much shorter, no longer than one month.  The anomalies in initial
soil moisture effect are mostly confined to the near-surface climate and somewhat in the upper
troposphere.
**3.2. Influence on temperature and other surface fluxes.**
The spatial distribution of averaged temperature (°C) from observations CRU (Fig.10 a and d)
and GTS (Fig.10 b and e) for JJAS 2003 and JJAS 2004 and their corresponding simulated from
control experiments (Fig.10 c and f) initialized with reanalysis soil moisture ERA20C is shown





in Fig.10. Table 2 resumes the PCC and the MB between the simulation of the temperatures
and CRU observation, calculated for west Sahel, central Sahel, Guinea coast and the whole
West African domain.
The CRU temperature displays a zonal distribution over the whole West Africa domain. The
maximum values about 34 °C are located over the Sahara, and the lowest temperatures are
found on the Guinea coast especially over the orographic regions such as Guinean highlands,
Cameroon mountains  and the Jos Plateau, where the temperature not exceeding 26°C. The two
observation datasets GTS and CRU are similar at large spatial scale with PCC about 0.99 over
the entire West African domain for both JJAS 2003 and JJAS 2004 (Table 2). However, the
extension and the amplitude of these maxima and minima are quite different in the two sets of
gridded observations.  While GTS (Fig.10b and e) observation displays large (small) areas with
maxima (minima) values, CRU (Fig.10a and d) has small (large) area of these maxima (minima)
values. The strongest mean warm biases between the two observation products, about 0.54°C
and 0.67°C respectively for JJAS 2003 and JJAS 2004, are located over the west Sahel sub-
region as compared to the other sub-regions (Table 2). The control experiments (Fig.10 c and
f) show a good agreement in representing the large-scale pattern of the observed temperature
(CRU) with PCC 0.99 for both JJAS 2003 and JJAS 2004 (Table 2), comprising the zone of the
meridional gradient of the surface temperature between Sahara Desert and Guinea coast which
is crucial for the African easterly jet (AEJ) evolution and formation (Thorncroft and Blackburn
1999; Cook 1999). However, some biases are noted at the local scale. The spatial extent of
temperature maxima and minima are well reproduced by control experiments, however their
magnitude are overestimate. The strongest warm mean biases of control experiments with
respect to CRU observation are about 2.68 °C and 2.14 °C respectively for JJAS 2003 and JJAS
2004, are found over the West Sahel sub-region (Table 2).
Figure 11 shows changes in mean temperature (°C) for JJAS 2003 and JJAS 2004, from dry
and wet experiments with respect to their corresponding control experiments. The dotted area
shows changes with that are statistical significance of 0.05 level. In the dry experiments, for
both JJAS 2003 and JJAS 2004, the dominant warm changes are located most the area under
the latitude 13°N, with maximum values located over the Guinea coast. However, a mixture of
warm and cool changes is located over the latitude 13°N (Fig.11a and c). On the other hand, for
the wet experiments, we found a dominant cool change over most of the area under the latitude
15°N, with maximum values around the latitude 13°N. While, a dominant warm change is
located above the latitude 15°N (Fig.11 b and d). Overall, the temperature is more sensitive to
soil moisture anomalies than precipitation over most of the domains studied.





For a better quantitative evaluation, the PDF distributions of the changes in mean temperature
in JJAS 2003 and JJAS 2004, over (a) central Sahel, (b) West Sahel, (c) Guinea and (d) West
Africa derived from dry and wet experiments compared to the corresponding control
experiments are shown in Figure 12. The temperature impact is homogeneous over central Sahel
and Guinea coast (Fig.12a and c). The strongest homogeneous impacts on temperature  of the
anomalies in initial soil moisture is located over the central Sahel, i.e, the dry (wet)  experiments
display a decrease (an increase) in temperature change with the peak mode of change at -1.5 °C
(0.5°C) as compared to other sub-regions, for both JJAS 2003 and JJAS 2004. Over the west
Sahel, both wet and dry experiments lead to a decrease of temperature. The impact on
temperature of the anomalies in soil moisture is somewhat sensitive to the wet and dry year, as
mentioned above, it is indicated by the lag between wet and dry experiments (Fig. 12). The
impact on dry and wet years depends on the area and the type of experience dry or wet. Overall,
the dry (wet) sensitivity experiments for 2m-temperature show a dominant increase (decrease)
of warming (cooling) for both JJAS 2003 and JJAS 2004 over most of the domains studied
except west Sahel where both dry and wet experiments lead to an increase of temperature
(Fig.12).
We now analyze the influence of initial soil moisture anomalies on land energy balance,
particularly on the surface fluxes sensible and latent heat. Figure 13 shows changes in sensible
heat fluxes (in W.m$^{-2}$) for JJAS 2003 and JJAS 2004, from dry and wet experiments compared
to their corresponding control experiments, and the dotted area shows changes with statistical
significance of 0.05 level. It can be seen in figure 13, the initial soil moisture anomalies strongly
affect the sensible fluxes.
In the dry experiments, the increase of sensible heat flux changes are located under the latitude
15°N, while the decrease of sensible heat flux changes are found over most of area above the
latitude 15°N for both JJAS 2003 and JJAS 2004 (Fig.13a, c). Conversely, in wet experiments
we have a dominant decrease of sensible heat flux changes, found over almost whole West
Africa domain except the orographic and somewhat along Guinea coastline for both JJAS 2003
and JJAS 2004 (see Fig 13b, d).
The PDF distributions of the change in sensible heat flux are displayed in Figure 14. The dry
(wet) experiments show an increase (decrease) of the sensible flux in both JJAS 2003 and JJAS
2004 over all the domains studied (Fig. 14). The impact in wet experiments is strong compared
to the dry experiments over central and west Sahel except over Guinea coast (Fig. 14). Overall,
the impact on sensible heat flux of soil moisture anomalies is homogeneous, i.e., dry
experiments tend to cause an increase of sensible heat flux while the wet experiments tend to





favor a decrease of sensible heat flux over all domains studied. The strongest impacts on
sensible heat flux in wet and dry experiments are found over respectively west Sahel and Guinea
coast, with peak modes about respectively -40W.m$^{-2}$ and 10W.mª (resp. Fig. 14,b and Fig. 14c).
Unlike to sensible heat case, changes in latent heat show opposite patterns, we found a dominant
decrease (increase) of latent heat flux in dry (wet) experiment over almost the studied domains.
Nevertheless, in the dry experiments, we found a sparse increase of latent heat over the Sahara
and Senegal (Fig.15b, d), while in wet experiment a sparse decrease is located over the Guinea
coast (Fig.15b, d). The PDF distributions of latent heat flux change are shown in Figure 15. It
can be seen, the impact on latent heat flux of soil moisture anomalies is homogeneous, i.e. dry
experiments lead to a decreasing in latent heat flux while the wet experiments lead to an
increasing in the latent heat flux over most of the domains studied. The strongest impact on
latent heat flux in wet and dry experiments are found over respectively west Sahel and Guinea
coast with peaks mode at 40W.m$^{-2}$ and -15W.m$^{-2}$ (resp. Fig. 16 b and Fig. 16 c). Overall, the
impacts in wet experiments on latent and sensible heat flux are strong compared to the dry
experiments over most of the domains studied, except over Guinea coast (Fig. 16).
In order to know if most of the changes in energy go to evaporating water, or to heating the
environment, we analyze in Fig. 17 the changes in Bowen ratio for JJAS 2003 and JJAS 2004,
from dry and wet experiments with respect to their corresponding control experiments. The
dotted area displays differences with statistical significance of 0.05 level. The soil moisture
anomalies strongly affected the Bowen ratio. The dry experiments show a dominant increase of
evaporation energy (Bowen ratio value in the range [0,1]) under the latitude 15°N for both JJAS
2003 and JJAS 2004 (Fig.17a, c). However, above latitude 15° N we found mixture of increase
and decrease of energy for environment heating (Bowen ratio value more than ±1) (Fig.17a, c).
For the wet experiments (Fig.17b, d), we found a dominant decrease of energy for environment
heating above the latitude 14°N (Bowen ratio less than -1), while under this latitude, we found
a mixture of decrease and increase of evaporation energy (Bowen ratio in the range [-1; 1]). As
expected, the areas where most of the changes in energy go to evaporating water are generally
coincident with temperature changes. The decrease (increase) in evaporation area coincides
with decrease (increase) of temperature change.
For a quantitative evaluation, the PDF distribution of the Bowen ratio is shown in Figure 18.
Over the Guinea coast, for both dry and wet experiments, most of energy go to evaporation with
decrease in Bowen ratio about, the dry (wet) experiments show an increase (decrease) of water
evaporation energy about 0.12 (-0.1) in both JJAS 2003 and JJAS 2004 (Fig. 18c).





On other hand, over the central Sahel (Fig. 18 a), for the dry and wet experiments, most of
energy goes to evaporation. The dry (wet) experiments more increase (decrease) the
evaporation energy with pic at 0.4 (-0.7) for both JJAS 2003 and JJAS 2004 over the central.
In contrary over west Sahel (Fig. 18b), in wet (dry) experiments, most of the energy goes to
heat the environment (to evaporation) with a decrease in Bowen ratio about -3 (-0.1).
We now examine the impact on the stability of planetary boundary layer (PBL) of the anomalies
in initial soil moisture. Soil moisture can influence rainfall by limiting evapotranspiration,
which affects the development of the daytime planetary boundary layer and thereby the
initiation and intensity of convective precipitation (Eltahir, 1998). Figure 19 shows the changes
in PBL (in m) for JJAS 2003 and JJAS 2004, from dry and wet experiments with respect to
their corresponding control experiments with dotted areas with statistical significance of 0.05
level. The soil moisture anomalies impact significantly the PBL. The dry experiments show an
increase of the PBL under the latitude 15°N, except a western part of west Sahel, while a
dominant decrease of PBL is shown above this latitude for both JJAS 2003 and JJAS 2004
(resp. Fig.19 a and c). For the wet experiments, a decrease of PBL is located over most of the
domains studied, however a sparse increase is found above the latitude 15°N. The PDF
distribution of PBL changes, computed over the area indicated in Figure1 is shown in Figure
20.   The impact on PBL is homogeneous over most of the domains studied (Fig.20). The dry
(wet) experiments lead to an increase (decrease) of PBL for both JJAS 2003 and JJAS 2004
over most of the domains studied. The strongest impacts on PBL, in the wet and dry
experiments, are found over respectively the west Sahel and Guinea coast, about respectively -
300 m and 150m. There is a dry (wet) air above the area  where there is increase (decrease) of
PBL, which results in warm (cool) and dry (moist) over most of the domains studied (see Fig.
7 and Fig. 8). These results are consistent with the work of Han and Pan 2003.
Summarizing the results of this section, simultaneously cooling (warming) of surface
temperature (wet experiments) should be associated with a smaller (greater) sensible heat flux,
greater (smaller) of latent heat and a smaller (greater) depth of the boundary layer over most of
domains studied.  These results are consistent with previous work of Eltahir and al. (1998).
Furthermore, sensible and latent heat fluxes, Bowen ratio and PBL responses to the anomalies
in initial soil moisture are somewhat sensitive to the contrast of year and experiments (wet and
dry).



## 4. Summary and conclusion

The impact of the anomalies in initial soil moisture on the subsequent summer mean climate over West Africa is explored using the RegCM4-CLM45. The results were established for two summers, JJAS 2003 (wet year) and JJAS 2004 (dry year). The sensitivity studies have been carried out on the West African domain, at a spatial resolution of 25 km × 25 km. The control runs are initialized by the reanalysis soil moisture ERA20C. We initialized the initial soil moisture at the wilting points and the field capacity for dry and wet experiments respectively.

The RegCM4.7 responses for precipitation simulation related to the anomalies in initial soil moisture anomalies show a homogeneous impact in the transition zones with a climate between wet and dry climate regimes over the central Sahel, i.e. dry (wet) experiments leading to precipitation decreasing (increasing). The strongest impact on precipitation in wet and dry experiments is found respectively over west and central Sahel, and the peak mode of change is about respectively 40% and -8%. The impact on precipitation in wet experiment is strong than in dry experiment. It is worth to note that, over the West Sahel and Guinea coast, both dry and wet experiments tend to cause a dominant increase of precipitation. This indicates that the increase of precipitation is more likely to happen not only in the wet experiments but also in the dry experiments. However, the increase of precipitation shown in the dry experiments results from the bringing of moistening from the ocean to the west Sahel and Guinea coast, an indication that the internal physics of the regional model is important in determining the model's surface climate. The soil moisture anomalies can persist up to three or even four months, however the significant response of precipitation to the anomalies in initial soil moisture is much shorter, no longer than one month. The effect of soil moisture anomalies is mostly confined to the near-surface climate and somewhat in the upper troposphere.

The temperature is more sensitive to the anomalies in initial soil moisture as compared to precipitation over most of the domains studied. The dry (wet) experiments for 2m-temperature show a dominant increase (decrease) of warming (cooling) for both JJAS 2003 and JJAS 2004 over most of the domains studied. The strongest homogeneous impacts of initial soil moisture anomalies on temperature is located over the central Sahel, i.e. the dry (wet) experiments display a decrease (an increase) in temperature change with the peak mode of change at -1.5 °C (0.5°C) as compared to other sub-regions, for both JJAS 2003 and JJAS 2004.



The impact on sensible and latent heat, the Bowen ratio and the PBL height of the anomalies in
initial soil moisture have been investigated in this study. We found that, simultaneously cooling
(warming) of surface temperature (wet experiments) is associated with a smaller (greater)
sensible heat flux, greater (smaller) of latent heat and a smaller (greater) depth of the boundary
layer over most of domains studied, with different magnitudes varying from one sub-region to
another. The strongest impacts on sensible heat in wet and dry experiments are found over
respectively west Sahel and Guinea coast, with peak modes about respectively $-40W.m^{-2}$ and
$10W.m^{-2}$ (resp. Fig. 14b and 14c). For latent heat, they are found over respectively west Sahel
and Guinea coast with peaks mode at $40W.m^{-2}$ and $-15W.m^{-2}$ (resp. Fig. 16b and 16c).  In wet
and dry experiments, the major impacts are found for the height of the PBL, over respectively
the west Sahel and Guinea coast, and about $-300$ m and 150m, respectively.
Furthermore, the impact of the anomalies in initial soil moisture on the Bowen ratio was
investigated to know if most of the changes in energy go to evaporating water, or to heating the
environment.
As expected, over the Guinea coast most of the changes in energy go to evaporation, the dry
(wet) experiments show an increase (decrease) of water evaporation energy in both JJAS 2003
and JJAS 2004. On other hand, over the central and west Sahel, for the dry and wet experiments,
most changes in energy respectively go to evaporation and to heat the environment.
We recognize that sensitivity of "dry" and "wet" experiments of initial soil moisture conducted
in this work, as in previous studies, were not supposed to simulate real climate since such
extremes are not current. However, these experiments can supply some estimation of the limits
of internal influence of soil moisture forcing. . To more complete this study, we will explore
the influence of the anomalies in initial soil moisture on climate extreme.
**Author contribution**
The authors declare to have no conflict of interest with this work. B. Koné and A. Diedhiou
fixed the analysis framework. B. Koné carried out all the simulations and figures production
according to the outline proposed by A. Diedhiou. B. Koné and A. Diedhiou, S. Anquetin and
A. Diawara worked on the analyses. All authors contributed to the drafting of this manuscript.
**Acknowledgements**
The research leading to this publication is co-funded by the NERC/DFID "Future Climate for
Africa" programme under the AMMA-2050 project, grant number NE/M019969/1 and by
IRD (Institut de Recherche pour le Développement; France) grant number UMR IGE
Imputation 252RA5.





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



**Tables and figures:**

| | Central Sahel | | West Sahel | | Guinea | | West Africa | |
|---|---|---|---|---|---|---|---|---|
| | PCC | MB (%) | PCC | MB (%) | PCC | MB (%) | PCC | MB (%) |
| **TRMM 2003** | 0.98 | 7.60 | 0.96 | -945 | 0.98 | -15.45 | 0.97 | -0.57 |
| **CTRL_2003** | 0.98 | -47.97 | 0.87 | -75.76 | 0.82 | -47.12 | 0.73 | -49.31 |
| **TMM 2004** | 0.98 | -0.62 | 0.99 | -7.03 | 0.98 | -16.96 | 0.97 | -1.56 |
| **CTRL_2004** | 0.98 | -47.89 | 0.87 | -68.35 | 0.85 | -51.97 | 0.77 | -50.56 |


**Table1:** The pattern correlation coefficient (PCC) and the mean bias (MB) for JJAS
precipitation for model simulation and observation TRMM with respect to CHIRPS, calculated
for Guinea coast, central Sahel, west Sahel and the entire West African domain during the
period 2003 and 2004.















| | Central Sahel | | West Sahel | | Guinea | | West Africa | |
|---|---|---|---|---|---|---|---|---|
| | PCC | MB (°C) | PCC | MB (°C) | PCC | MB (°C) | PCC | MB (°C) |
| GTS 2003 | 0.99 | 0.31 | 0.99 | 0.54 | 0.99 | 0.28 | 0.99 | 0.39 |
| CTRL_2003 | 0.99 | 1.52 | 0.99 | 2.68 | 0.99 | -0.34 | 0.99 | 0.85 |
| GTS 2004 | 0.99 | 0.32 | 0.99 | 0.67 | 0.99 | 0.28 | 0.99 | 0.40 |
| CTRL_2004 | 0.99 | 1.50 | 0.99 | 2.14 | 0.99 | -0.57 | 0.99 | 0.51 |


**Table2:** The pattern correlation coefficient (PCC) and the mean bias (MB) for JJAS 2m-
temperature for model simulation and observation (GTS) with respect to CRU, calculated for
Guinea coast, central Sahel, west Sahel and the entire West African domain during the period
2003 and 2004.














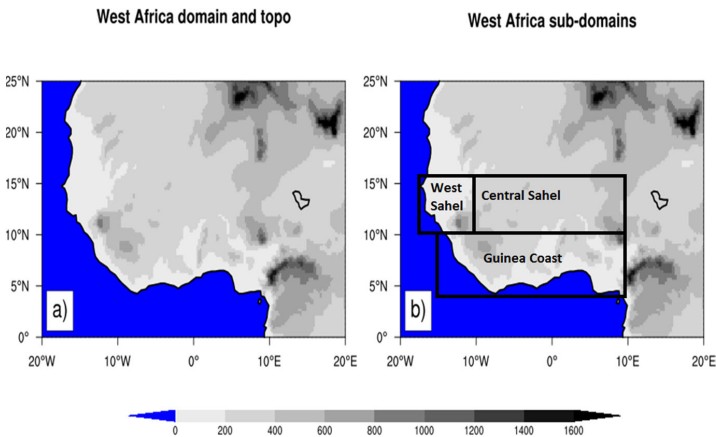



**Figure 1:** Topography of the West African domain. The analysis of the model result has an emphasis on the whole West African domain and the three subregions Guinea coast, central Sahel and west Sahel, which are marked with black boxes.























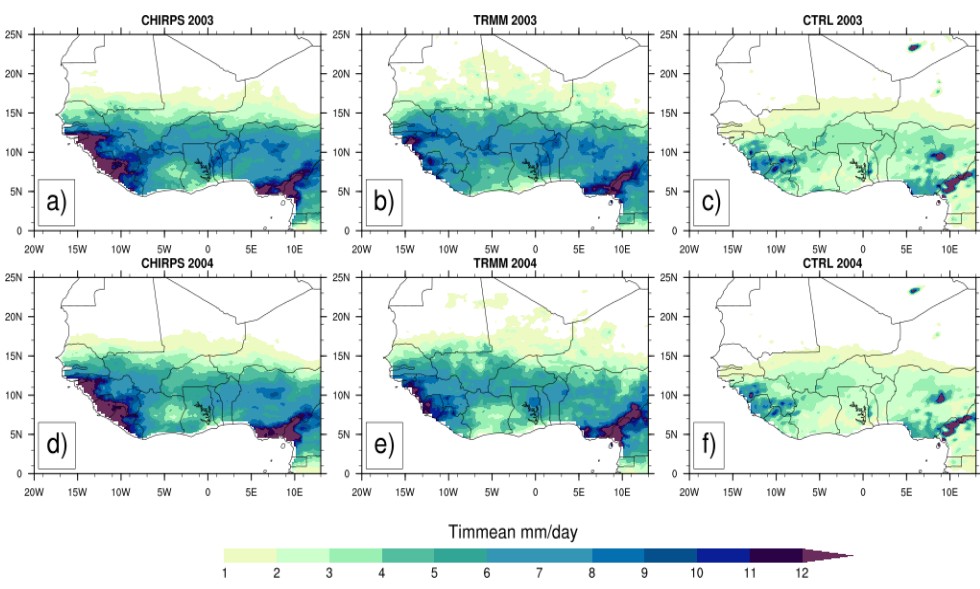


**Figure2:** Observed 4-month averaged (JJAS) precipitation (mm/day) from CHIRPS (a and d)
and TRMM (b and e) for 2003 and 2004 and their corresponding simulated control experiments
(CTRL) (c and f) with the reanalysis initial soil moisture ERA20C.

















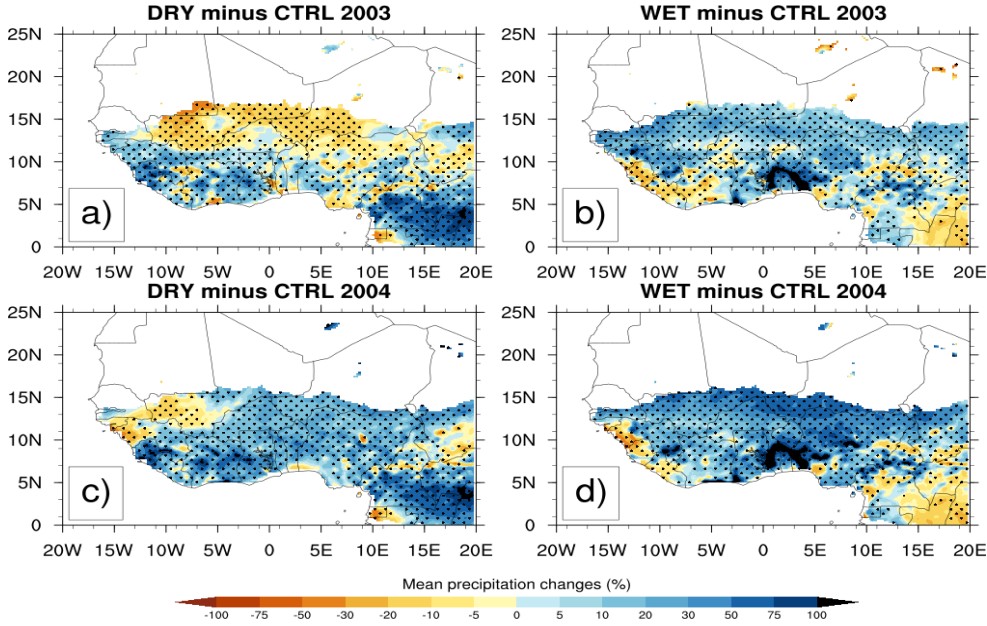



**Figure3:** Changes in mean precipitation (in %) for JJAS 2003 and JJAS 2004, from dry (resp. a and c) and wet (resp. b and d) experiments with respect to their corresponding control experiment, the dotted area shows differences that are statistically significant at 0.05 level.















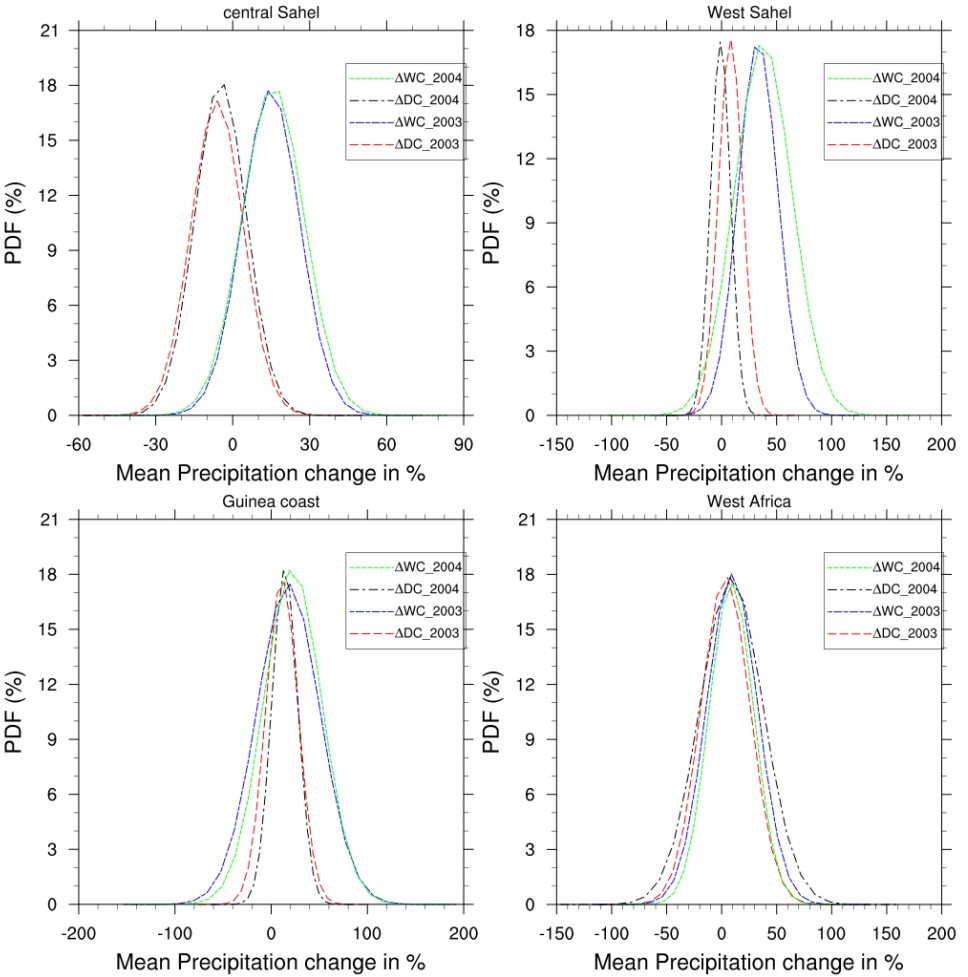

**Figure 4:** PDF distributions (%) of mean precipitation changes in JJAS 2003 and JJAS 2004, over (a) central Sahel, (b) West Sahel, (c) Guinea and (d) West Africa derived from dry (DC) and wet (WC) experiments compared to their corresponding control experiment.






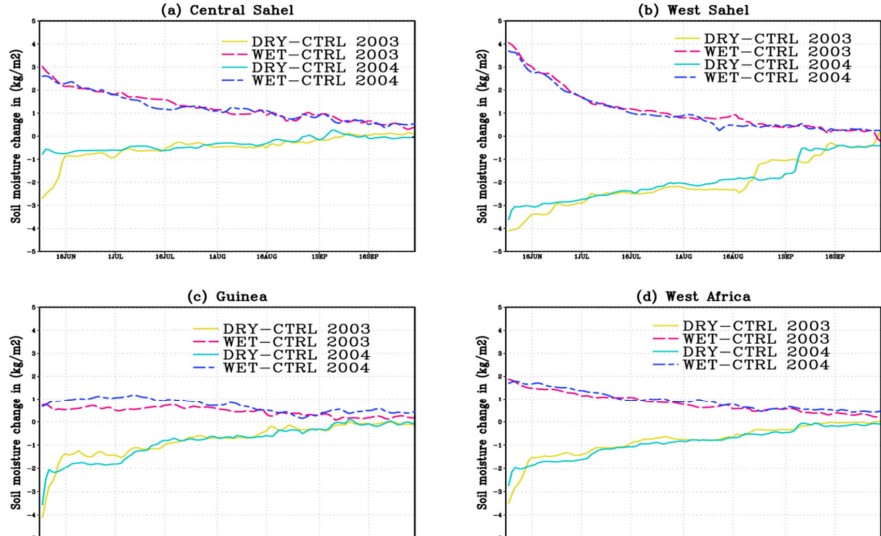


**Figure 5:** Daily domain-average soil moisture changes for JJAS 2003 and JJAS 2004, from dry
(a and c) and wet (b and d) experiments with respect to their corresponding control experiment.






















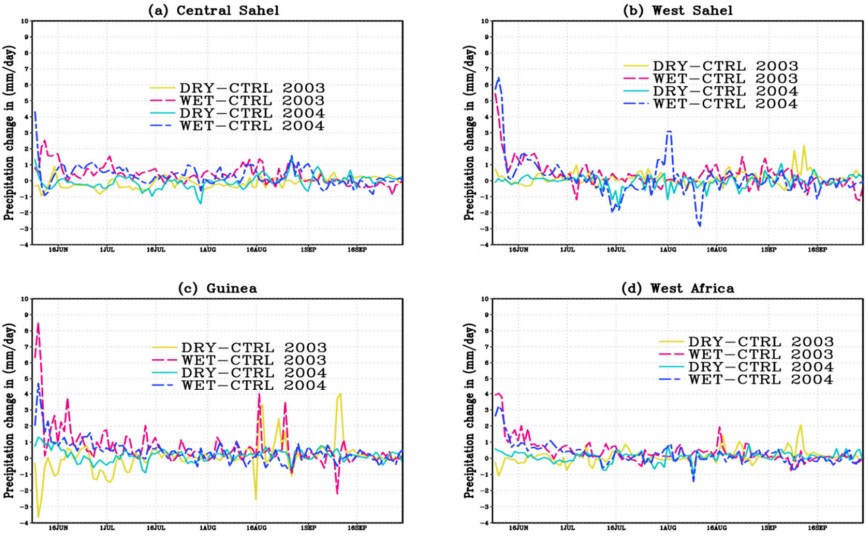



**Figure 6:** Daily domain-average precipitation changes for JJAS 2003 and JJAS 2004, from dry
(a and c) and wet (b and d) experiments with respect to their corresponding control experiment.


















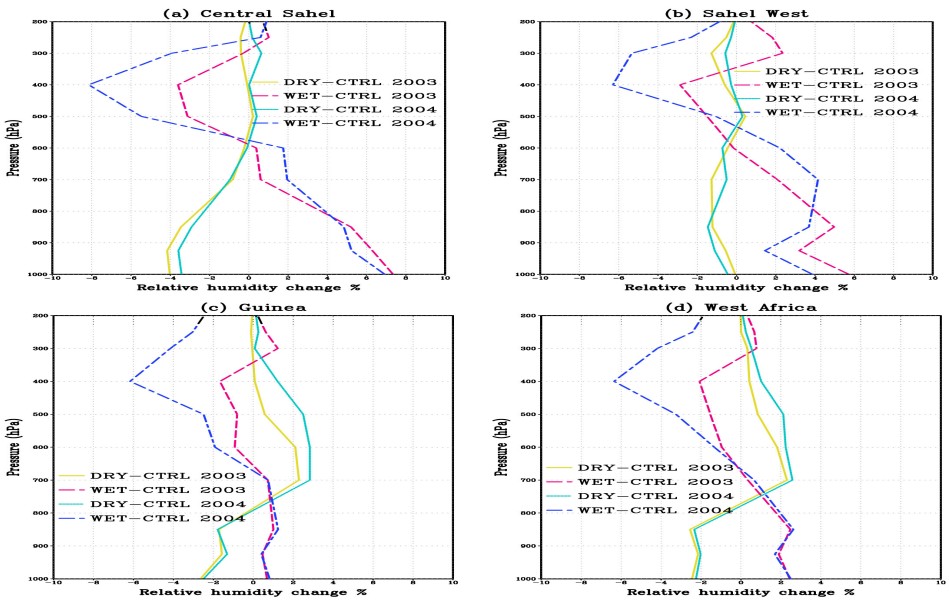

**Figure 7:** Vertical profile changes in humidity for JJAS 2003 and JJAS 2004 from the dry and wet experiments with respect to corresponding control experiment over (a) central Sahel, (b) west Sahel, (c) Guinea coast, and (d) West Africa.





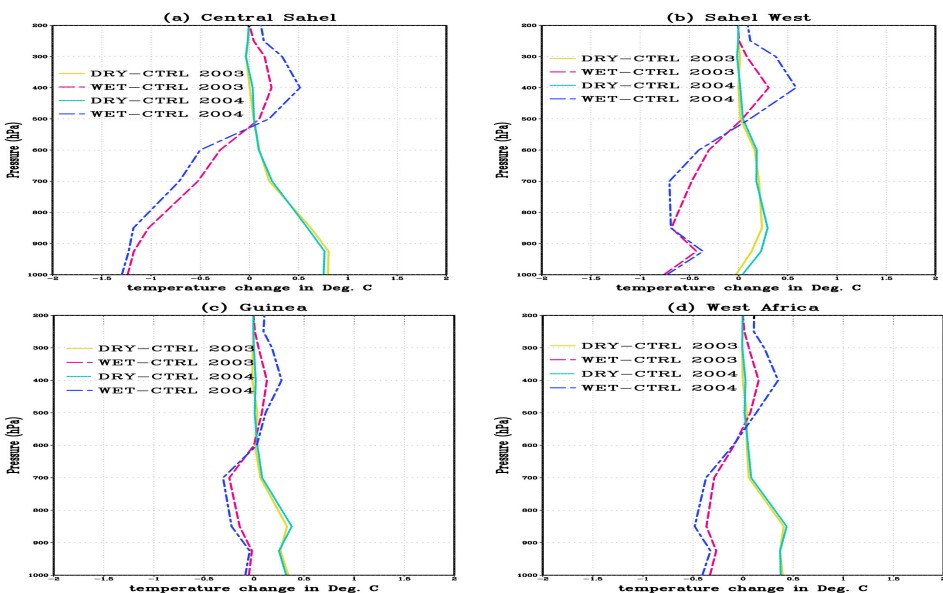

**Figure 8:** Vertical profile changes in temperature for JJAS 2003 and JJAS 2004 from the dry and wet experiments with respect to their corresponding control experiment over (a) central Sahel, (b) west Sahel, (c) Guinea coast, and (d) West Africa.



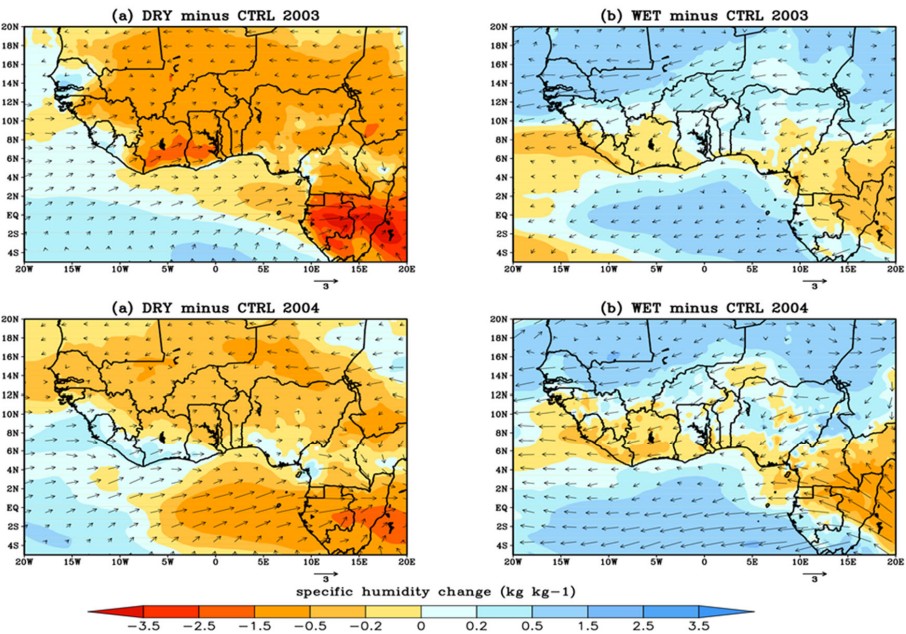

**Figure 9:** The lower tropospheric wind (850hpa) and moisture bias for JJAS 2003 and JJAS 2004 from the dry (a and c) and wet (b and d) experiments with respect to their corresponding control experiment.


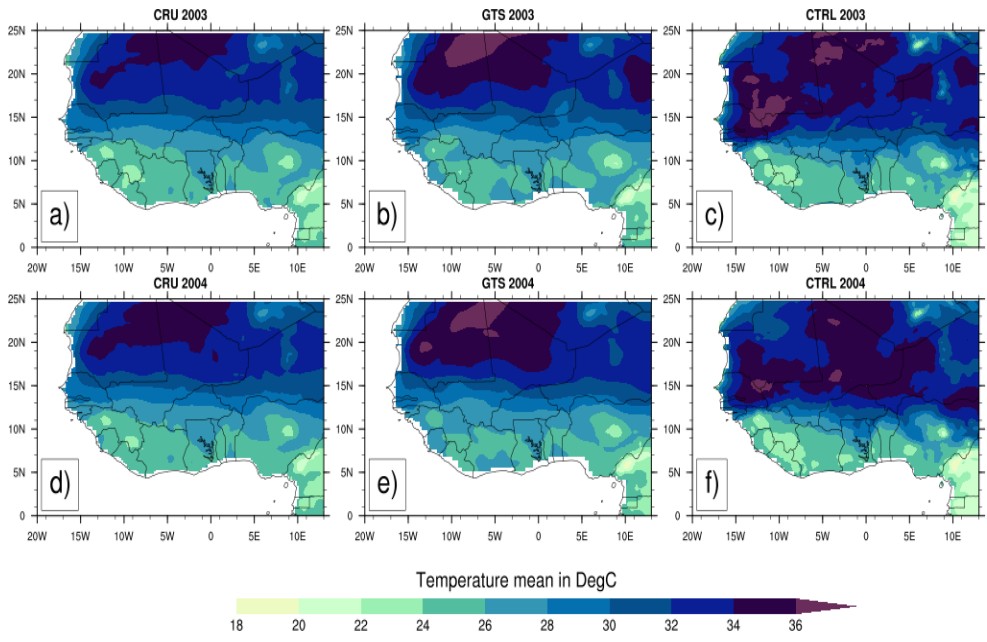


**Figure 10:** Observed 4-month averaged (JJAS) 2m-temperature (°C) from CRU (a and d) and
GTS (b and e) for  2003 and 2004 and their corresponding simulated control experiment (c and
f) with the reanalysis initial soil moisture ERA20C.

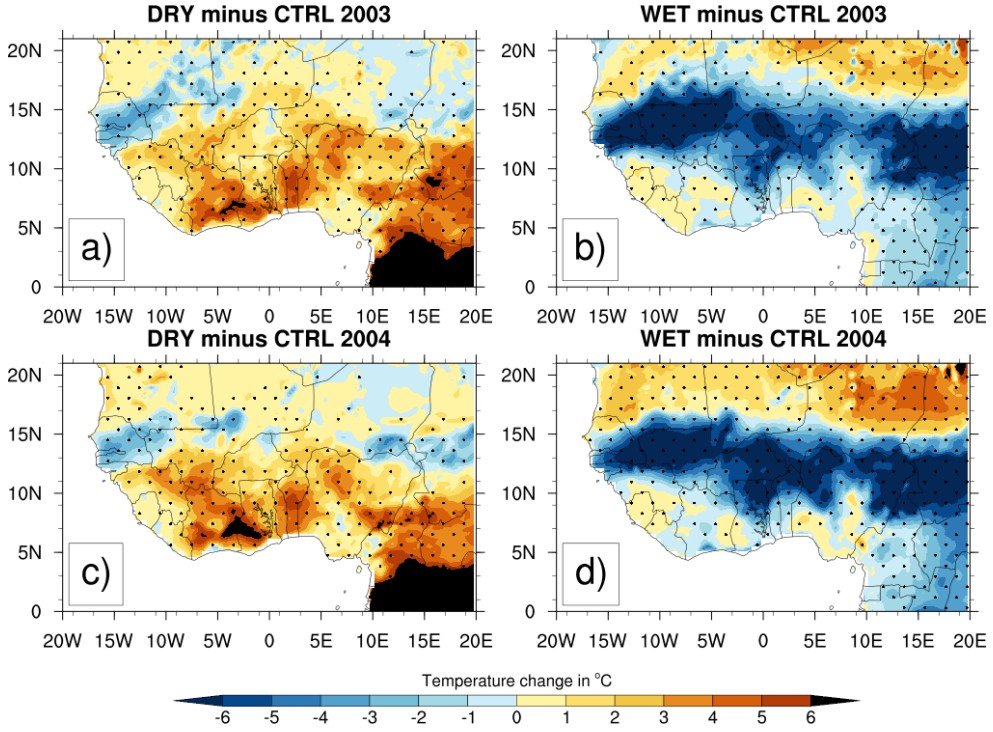


**Figure 11:** Changes in 2m-temperature (°C) for JJAS 2003 and JJAS 2004, from dry (resp. a
and c) and wet (resp. b and d) experiments with respect to their corresponding control
experiment, the dotted area shows differences that are statistically significant at 0.05 level.




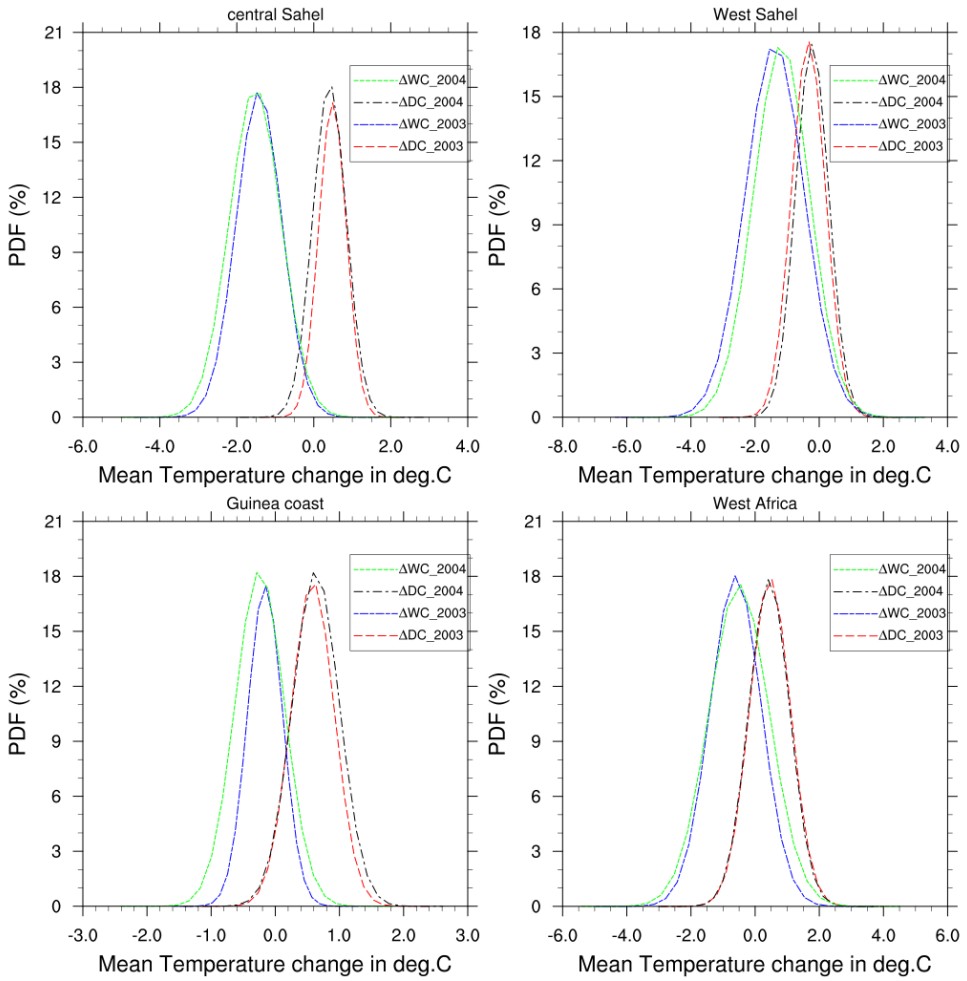

**Figure 12:** PDF distributions (%) of mean temperature changes in JJAS 2003 and JJAS 2004, over (a) central Sahel , (b) West Sahel, (c) Guinea and (d) West Africa derived from dry and wet experiments compared to their corresponding control experiment.





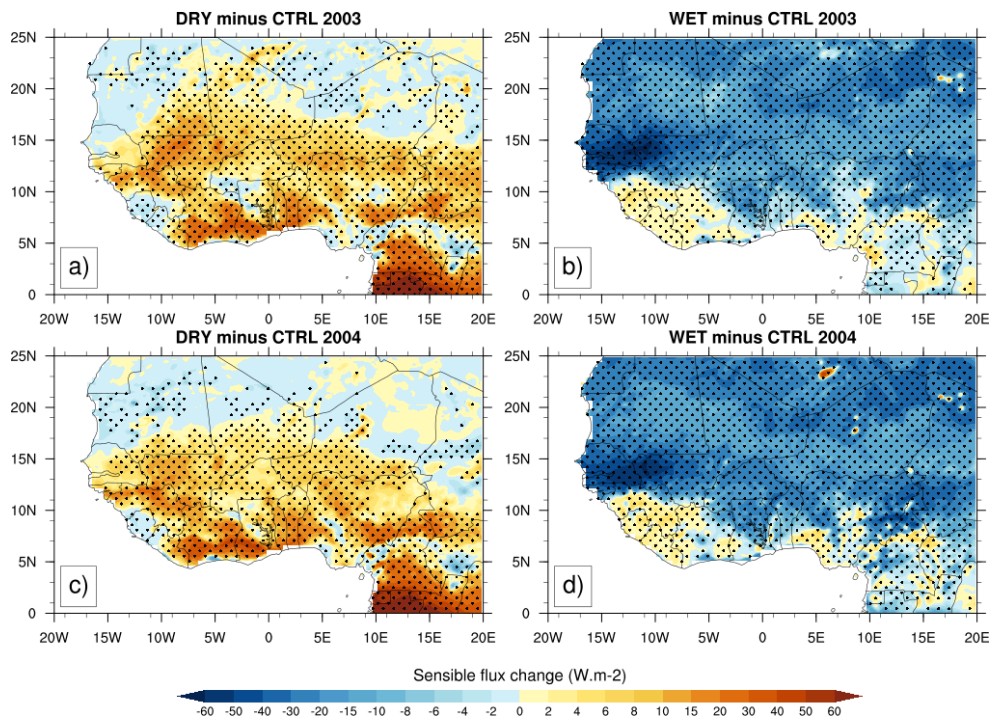



**Figure 13:** Same as Fig.11 but for sensible heat fluxes (in W.m⁻²).



















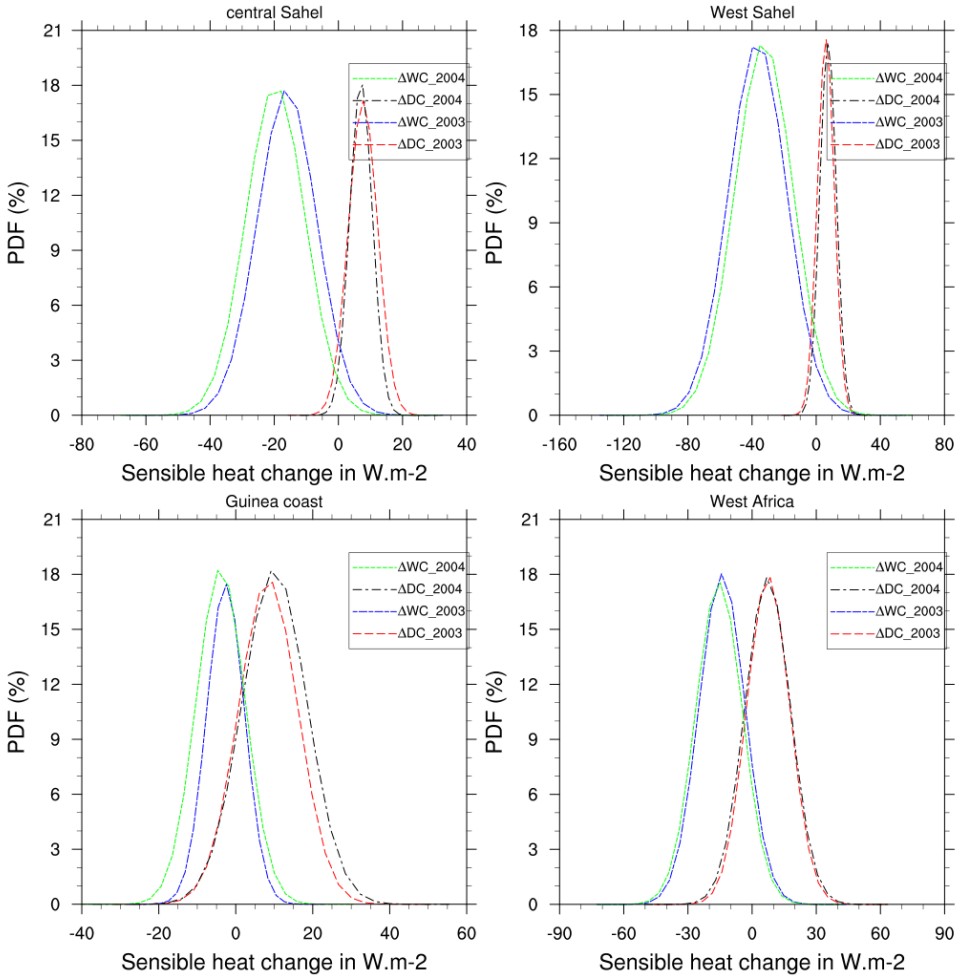



**Figure 14:** Same as Fig.12 but for sensible heat fluxes (in W.m$^{-2}$).











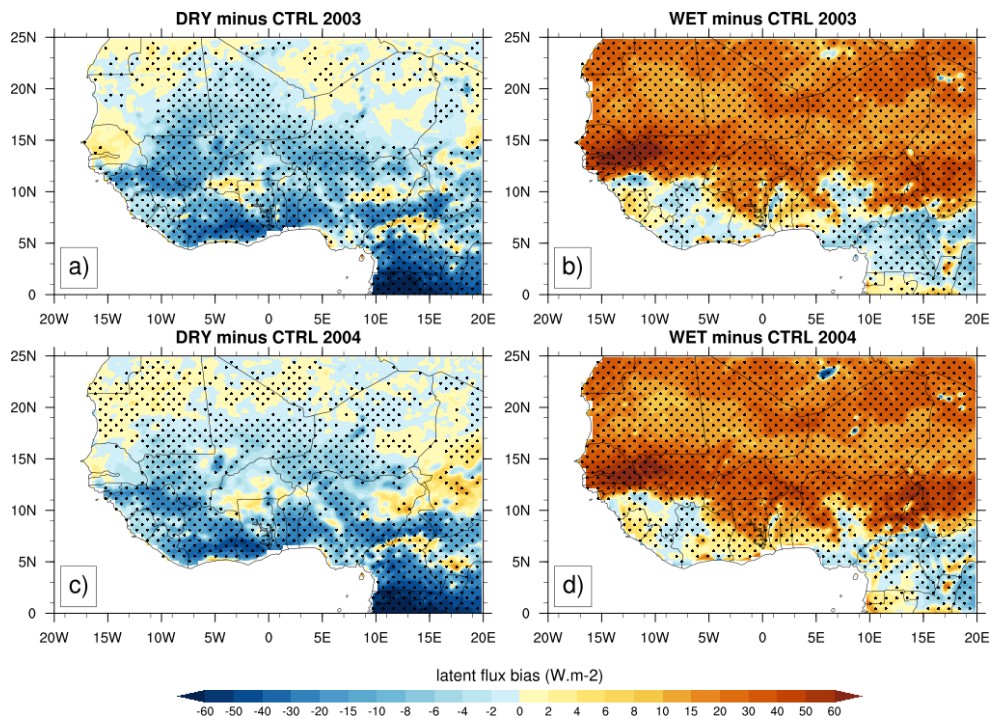



**Figure 15:** Same as Fig.11 but for latent heat fluxes (in W.m⁻²).




















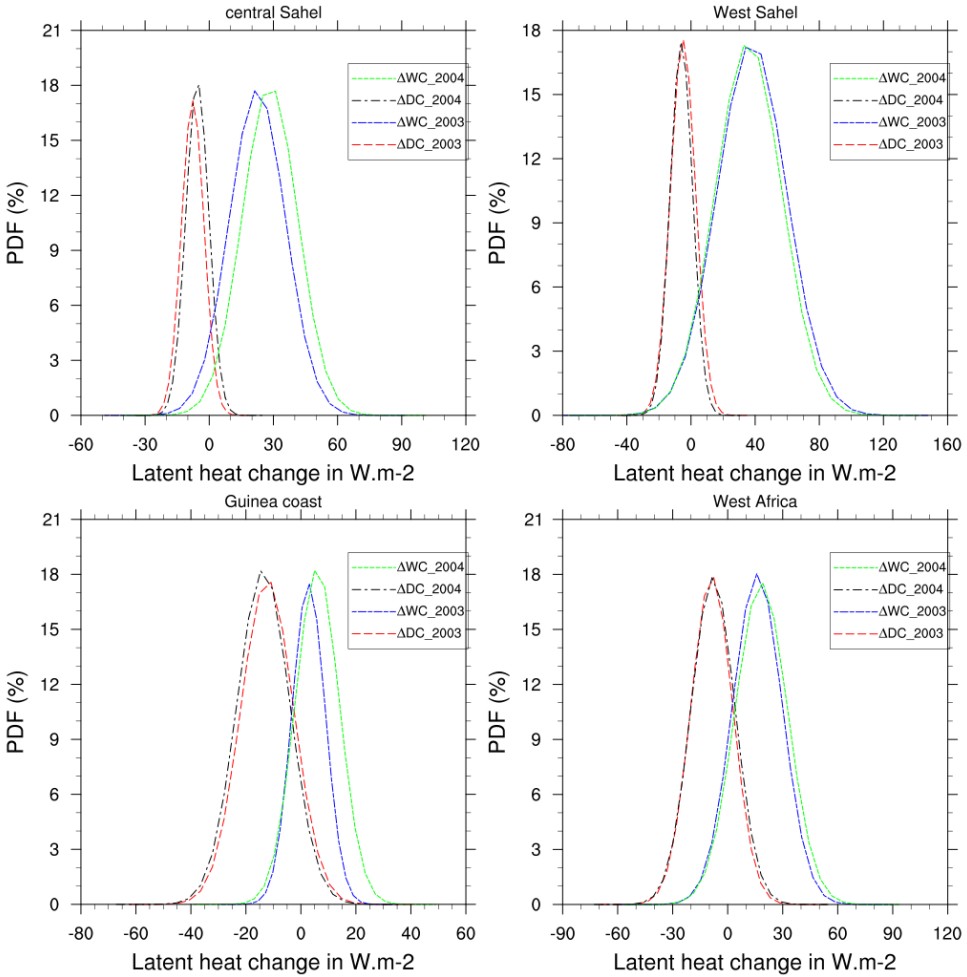

**Figure 16:** Same as Fig.12 but for latent heat fluxes (in W.m$^{-2}$).





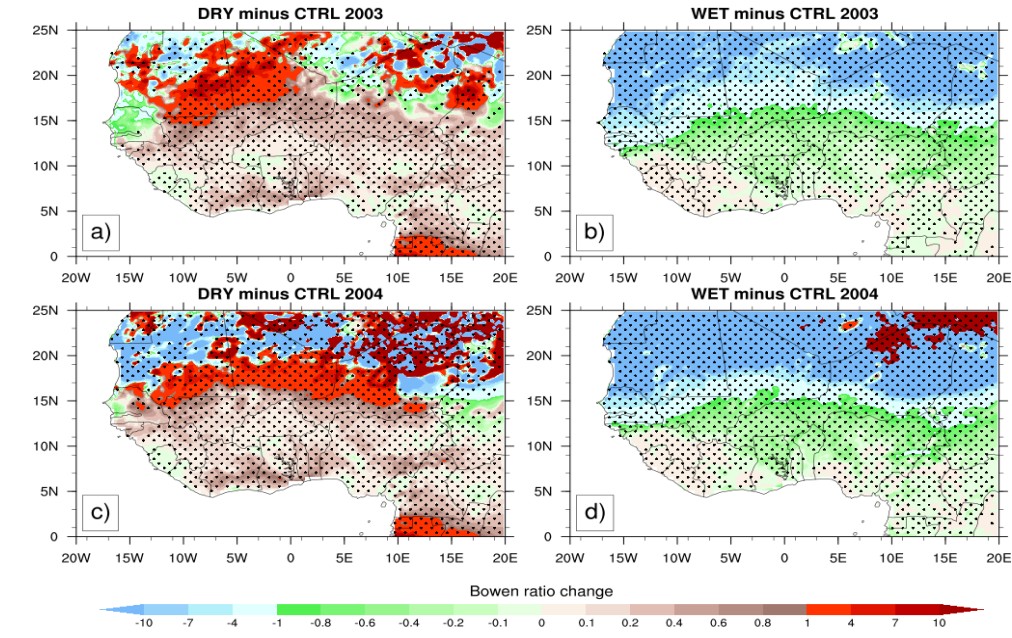



**Figure 17:** Same as Fig.11 but for Bowen ratio.




















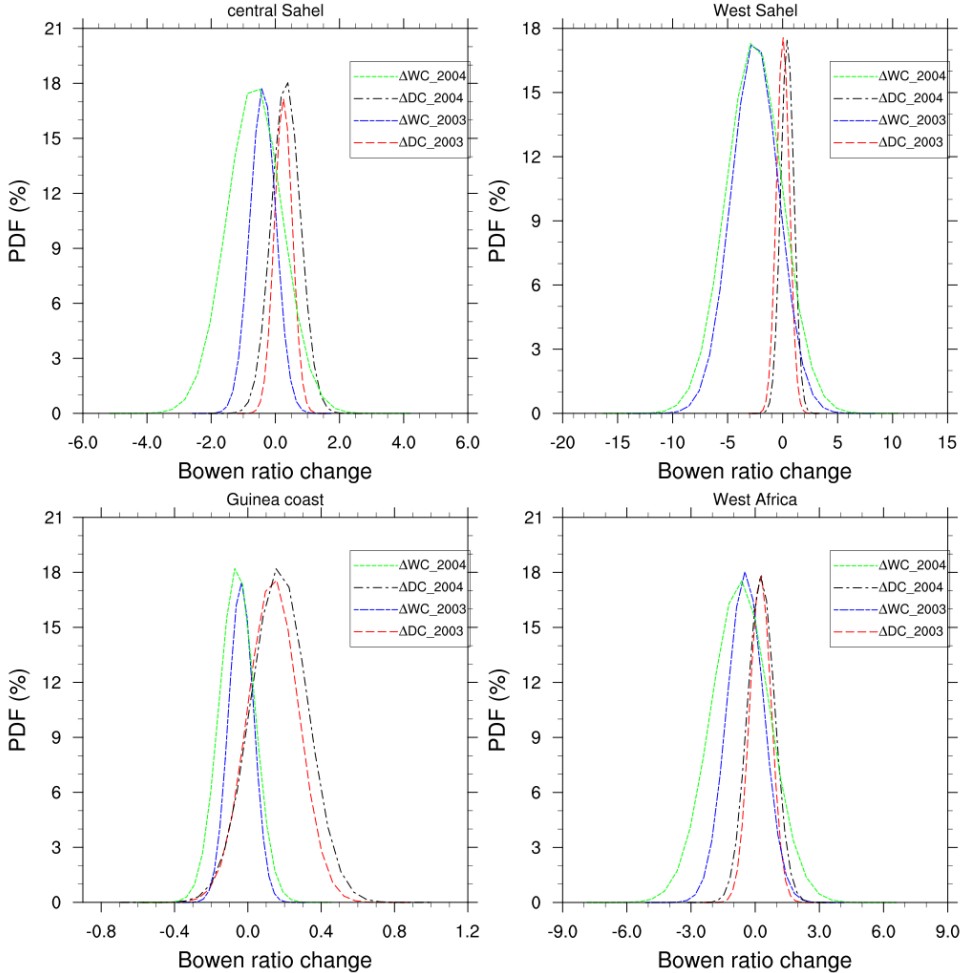



**Figure 18:** Same as Fig.12 but for Bowen ratio.














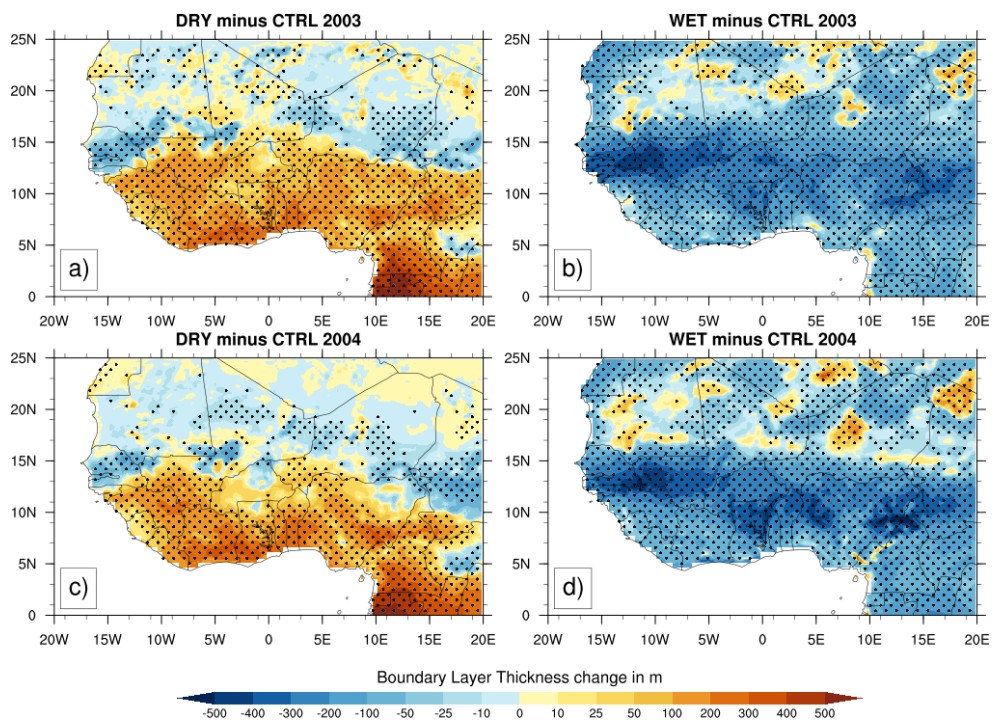

**Figure 19:** Same as Fig.11 but for the change of the height of the planetary boundary layer (in m).



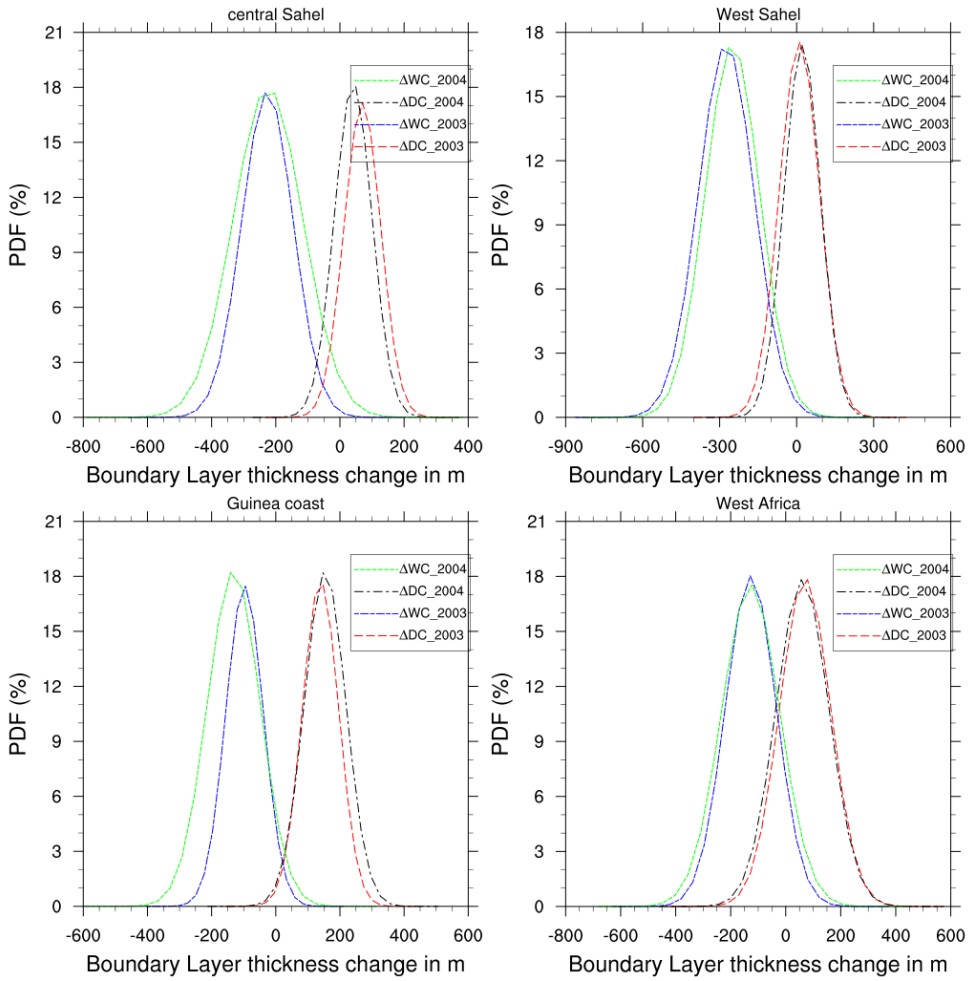



**Figure 20:** Same as Fig.12 but for the height of the planetary boundary layer (in m).





