# Peer review of "Influence of initial soil moisture in a Regional Climate Model study over West Africa. Part 1: Impact on the climate mean"

_Hydrology and Earth System Sciences, 2020_

## Referee Comment (RC1) · Anonymous Referee #1 · 17 Jun 2020

Reviewer Comments Influence of initial soil moisture in a Regional Climate Model study over West Africa: Part 1: Impact on the climate mean

The manuscript describes a range of experiments to determine the impact of soil moisture anomalies on the summer (JJAS) climate of West Africa. The experiments include 3 experiments for each JJAS 2003 (a wet year) and JJAS 2004 (a dry year). The 3 model experiments are a control run using ERA20C initial soil moisture, a dry initial soil moisture state, and a wet initial soil moisture state. The impacts of these varying initial states are discussed for each year, precipitation, temperature, sensible and latent heat, the Bowen ratio, and PBL height. For precipitation, the authors found that

(generally) results agree with previous studies and homogeneous impacts are found in the transition zones between wet and dry climate regimes, though the wet experiment impacts precipitation more strongly than dry. However, the results vary regionally, for example the West Sahel and Guniea coast show increased precipitation in both dry and wet experiments. Temperature showed more sensitivity to soil moisture initialization, where dry experiments caused increased warming and vice versa. Wet experiments caused cooling of surface temperatures, smaller sensible heat flux, greater latent heat flux, and a smaller depth of the PBL which was mostly homogeneous over the region. Overall the results may be significant for future studies of soil moisture initialization over West Africa.

I think that the manuscript shows merit, and soil moisture initialization remains important for climate prediction - the authors indeed show differences in precipitation and temperature depending on soil moisture initial state; however, I have some concerns about the experiment design. I'm curious about the choice of JJAS 2003 and 2004, and if these two years allow for a robust study of the influence of soil moisture initialization. I have a number of other comments below, but I believe that this paper needs major revisions before acceptance.

Major/Specific Comments: 1. As noted above, my main concern stems from your year choices. While this results in 6 experiments for comparison, I am not convinced that the results are robust given only a 2 year sample size. Moreover, I'm curious how these years were chosen - are they extreme wet and dry years? How often do years such as these occur? How is "wet" and "dry" defined?

2. I'm not sure I understand why you discard the first 7 days as spin-up - perhaps because I'm used to prediction, where those 7 days are included in the forecast and would show large impacts of soil moisture initialization.

- 3. Line 152: Is a normal distribution the correct choice here?
- 4. Line 156 157: Given you only have a few experiments, a significance test would
be difficult to achieve. Do you feel that you have sufficiently large "n" to address significance? You may have enough grid points to address spatial significance (given that you modify your n due to autocorrelation, but I note in my minor comments that you should include your sample size), but temporally I don't feel the results are robust.

5. It seems that in a broad sense your results largely agree with those from Koster et al., etc. which show that the largest impacts occur in transition zones, and you note that this is poorly understood over West Africa. I'd like to see some additional explanation on the impacts of this manuscript to the scientific community - how does this manuscript provide better understanding and what could this better understanding lead to? Basically, your conclusions are more of a summary/rehashing, and I'd like to see more impactful statements.

Minor/Technical Comments: 1. I noticed a number of grammatical and spelling errors in the manuscript, I suggest having someone read and edit the manuscript specifically for editorial remarks such as these.

2. You use a number of parenthetical references such as "impacts of the wet (dry) soil moisture on wet (dry) years etc." - I do not mind these at all, but sometimes the text is very difficult to read when they are used in excess. For example, line 464-468.

3. Define the lat and lon range of your domain(s).

4. You make a number of references to differences at the "local" scale - how do these differences impact your results, if at all.

5. Table 1: What is your sample size (how many grid points?)

6. Fig. 1 and your domains, did you only include land points for precipitation and temperature?

7. You may want to consider keeping a consistent naming scheme for your experiments - I found it hard to follow at times when you had wet vs. dry years, wet vs. dry initializations, etc.

HESSD

---

## Referee Comment (RC2) · Anonymous Referee #2 · 24 Jun 2020

In these two manuscripts Koné et al. initialize a regional climate model with different soil moisture conditions to study the effect of soil moisture on the mean climate and on climate extremes. I believe that these two manuscripts are interesting and potentially of value to the research community, but the authors do not sufficiently justify their analysis design nor do they contextualize their initial conditions relative to observed variability. I would urge the authors to consider providing more context to justify their choices to better help the reader interpret the results.

Choice of study years: How were the years 2003 and 2004 chosen? Perhaps show a time series of precipitation anomalies in your study region to highlight why you chose

2003 and 2004. "Dry" and "wet" are very subjective terms. Or consider showing observed precipitation and soil moisture anomalies during 2003 and 2004.

Choice of soil moisture data: Why initialize soil moisture with ERA 20C, which includes only surface forcings of surface pressure and marine winds only? How well does this dataset compare to satellite-derived observations? ERA 20C isn't really a reanalysis of soil moisture, because the soil moisture doesn't include any observations. If the authors want an observationally-based soil moisture dataset, they could consider a product like GLEAM (Martens et al., 2017; Miralles et al., 2011)

You show the observations for precipitation, perhaps show observations for soil moisture too. You could show ERA20C (which you added) and maybe GLEAM as an independent dataset

Using global wilting points and field capacity: In the dry and wet runs the authors use a uniform wilting point and field capacity everywhere. But we expect that both the permanent wilting point and the water holding capacity of the soil differs by location (see, for example Figure 6 of Leenaars et al., 2018). These two values will be radically different in the Sahara Desert and southern Nigeria, for example. The assumptions made here, that wilting point is 0 or that field capacity is 0.489 is likely unrealistic for many locations, and these extreme initial conditions may be affecting the results. The authors need to more completely justify the use of these initial conditions, as opposed to using the maximum and minimum observed values, for example.

Effect of methods on the analysis: The fact that the wet year (2003) and the dry year (2004) look the same in most graphs when used as the control seems to indicate that they're either quite similar, or that the values for initial soil moisture are incredibly strong, and are overwhelming everything about the dry vs wet year. Are extreme values of soil moisture like this really useful? If so, the authors need to better justify them. I understand that this is a sensitivity analysis, but the authors need to contextualize how relevant this sensitivity analysis is to the conditions of the real world.

Starting at a soil moisture of 0 is quite extreme. Show, for example the local minimum and maximum soil moisture estimated for the region in the target starting month (June) in an observational dataset as comparison. Showing local min/max for each pixel would demonstrate how the initial conditions used compare to what has historically been experienced.

References: Leenaars JG, Claessens L, Heuvelink GB, Hengl T, González MR, van Bussel LG, Guilpart N, Yang H, Cassman KG. Mapping rootable depth and root zone plant-available water holding capacity of the soil of sub-Saharan Africa. Geoderma. 2018 Aug 15;324:18-36.

Martens, B., Miralles, D.G., Lievens, H., van der Schalie, R., de Jeu, R.A.M., Fernández-Prieto, D., Beck, H.E., Dorigo, W.A., and Verhoest, N.E.C.: GLEAM v3: satellite-based land evaporation and root-zone soil moisture, Geoscientific Model Development, 10, 1903–1925, doi: 10.5194/gmd-10-1903-2017, 2017.

Miralles, D.G., Holmes, T.R.H., de Jeu, R.A.M., Gash, J.H., Meesters, A.G.C.A., Dolman, A.J.: Global land-surface evaporation estimated from satellite-based observations, Hydrology and Earth System Sciences, 15, 453–469, doi: 10.5194/hess-15-453-2011, 2011.

---

## Author Comment (AC1) · 4 Aug 2020

Reply to the comments of the referee 1 for the manuscript hess-2020-112

We thank the reviewers for the careful review and positive comments which helped to improve the manuscript. Please find our answers to comments in italic as well as suggested text changes in yellow in the revised version.

Introduction of the reviewer The manuscript describes a range of experiments to determine the impact of soil moisture anomalies on the summer (JJAS) climate of West Africa. The experiments include 3 experiments for each JJAS 2003 (a wet year) and

JJAS 2004 (a dry year). The 3 model experiments are a control run using ERA20C initial soil moisture, a dry initial soil moisture state, and a wet initial soil moisture state. The impacts of these varying initial states are discussed for each year, precipitation, temperature, sensible and latent heat, the Bowen ratio, and PBL height. For precipitation, the authors found that (generally) results agree with previous studies and homogeneous impacts are found in the transition zones between wet and dry climate regimes, though the wet experiment impacts precipitation more strongly than dry. However, the results vary regionally, for example the West Sahel and Guinea coast show increased precipitation in both dry and wet experiments. Temperature showed more sensitivity to soil moisture initialization, where dry experiments caused increased warming and vice versa. Wet experiments caused cooling of surface temperatures, smaller sensible heat flux, greater latent heat flux, and a smaller depth of the PBL which was mostly homogeneous over the region. Overall the results may be significant for future studies of soil moisture initialization over West Africa. I think that the manuscript shows merit, and soil moisture initialization remains important for climate prediction - the authors indeed show differences in precipitation and temperature depending on soil moisture initial state; however, I have some concerns about the experiment design. I'm curious about the choice of JJAS 2003 and 2004, and if these two years allow for a robust study of the influence of soil moisture initialization. I have a number of other comments below, but I believe that this paper needs major revisions before acceptance.

Major/Specific Comments: 1. Comments: As noted above, my main concern stems from your year choices. While this results in 6 experiments for comparison, I am not convinced that the results are robust given only a 2 year sample size. Moreover, I'm curious how these years were chosen -are they extreme wet and dry years? How often do years such as these occur? How is "wet" and "dry" defined?

Author's response: Thank you for your comment. The aim of this study is to investigate how soil moisture initialization at the beginning of the rainy season may impact the intraseasonal variability of temperature and precipitation mean and extremes within the

**HESSD**
subsequent season (June to September). For this purpose, we set up an ensemble of 3 experiments each with simulations starting from June 1st to September 30th. The difference between these 3 experiments is on the change of soil moisture initial condition during the first day of the simulation (June 1st): For each experiment, we applied (i) a reference initial soil moisture condition, (ii) then a wet initial soil moisture condition, and finally (iii) a dry initial soil moisture condition. As previous studies demonstrated that the impact of soil moisture initial condition is no longer than 4 months (120 days; Hong and Pan. (2000); Kim and Hong (2006); Koster et al. (2010); Santanello et al. (2019)), we decided to perform these 3 experiments (wet, dry and reference soil moisture initial conditions) during two contrasted years in West Africa (in 2003 wet year and in 2004 dry year compared to the mean 1950-2004 cf fig. below) in the aim to assess the results of the sensitivity study, whatever the year studied.

The figure below on the anomaly of rainfall in the Sahel from 1950 to 2004 is from http://research.jisao.washington.edu/data/sahel/022208/ and shows differences between wet and dry years. It helps to answer how these years (2003 and 2004) were chosen and how often those years occur. Sahel region as well as the West Africa region experienced a period of wet years in the 50-60s followed by a period of 30 dry years until 90s. Annual rainfalls during the last 30 years are highly variable and although the Sahel has seen a "recovery" in rainfall since the 1980s, cumulative precipitation has not returned to pre-1960s levels.

2. Comments: I'm not sure I understand why you discard the first 7 days as spin-up – perhaps because I'm used to prediction, where those 7 days are included in the forecast and would show large impacts of soil moisture initialization.

Author's response: Thank you very much. Spin-up is a concern when there is a lack of data or seasonal simulation (Rahman and Lu, 2015). Overestimating the spin-up period would lead to a loss of important information. Likewise, an underestimation will lead to integrate errors in the analysis due to the fact that the model does not reach the dynamical equilibrium between the lateral forcing and the internal physical dynamic of
the model. Yes, you're right, Anthes et al. (1989) demonstrated that regional models attain the dynamical equilibrium in 2-3 days spin-up period. However, Kang and al. (2014) by comparing different land surface schemes (BATS and CLM3) and different periods of spin-up to simulate June – July – August precipitations recommended 7 days as spin-up period. In this study, we used CLM4.5 as land surface scheme (Oleson et al., 2013) which has a more complex design. That's why we used 7 days as spin-up period.

3. Comments: Line 152: Is a normal distribution the correct choice here?

Author's response: Use of normal law in probability density functions (PDF) to explore the frequency and severity of climate extremes is not new ((Katz and Brown 1992; Kiktev et al. (2003), Alexander et al. (2006)).

Andronova and Schlesinger (2001) and Knutti et al. (2002) used this method to compute the biases and to represent the uncertainty in physical processes and feedbacks in climate models. Dessai et al (2005) used it to assess the uncertainty in regional climate change projections. Jaeger and Seneviratne (2011) used this method to assess the impact of soil moisture-atmosphere coupling on European climate extremes and trends in a regional climate model.

4. Comments: Line 156 - 157: Given you only have a few experiments, a significance test would be difficult to achieve. Do you feel that you have sufficiently large "n" to address significance? You may have enough grid points to address spatial significance (given that you modify your n due to autocorrelation, but I note in my minor comments that you should include your sample size), but temporally I don't feel the results are robust.

Author's response: Thank you for your comment. The number of our sample size (number of grid points) is 20748 (182x114) covering the entire West Africa region. The statistical significance is performed over these grid points. We test the null hypothesis that the sample means are from the same population (i.e. H0: ave1=ave2) after
computing temporal average and p-value at each grid point. We specify a critical significance level for the ttest and test if the means are different at each grid point. We used NCL to perform this significance test.

5. Comments : It seems that in a broad sense your results largely agree with those from Koster et al., etc. which show that the largest impacts occur in transition zones, and you note that this is poorly understood over West Africa. I'd like to see some additional explanation on the impacts of this manuscript to the scientific community - how does this manuscript provide better understanding and what could this better understanding lead to? Basically, your conclusions are more of a summary/rehashing, and I'd like to see more impact full statements.

Author's response: Thank you for your comment. The conclusion has been improved based on comments of reviewers. This paper (part 1) shows that sensitivity to soil moisture initial conditions is stronger with temperature than with precipitation and that the impacts are different depending the sub-regions. Precipitation and Temperature over West Sahel and Guinea Coast sub-regions closer to the Atlantic Ocean react differently to soil moisture initial conditions than over the drier region of Central Sahel.

Minor/Technical Comments:

1. Comments: I noticed a number of grammatical and spelling errors in the manuscript, I suggest having someone read and edit the manuscript specifically for editorial remarks such as these.

Author's response: Thank you for your comment. We did our best to improve the manuscript.

2. Comments: You use a number of parenthetical references such as "impacts of the wet (dry) soil moisture on wet (dry) years etc." - I do not mind these at all, but sometimes the text is very difficult to read when they are used in excess. For example, line 464-468.
Author's response: Thank you for your comment. We reduce this style of writing in the revised version and make it easier to read.

3. Comments from referee 1: Define the lat and lon range of your domain(s).

Author's response: West Africa simulation domain ć Number of grid points nlon\*nlat = 182x114 = 20748 ć Longitude range: 20°W to 19.82°E by 0.22 degrees resolution ć Latitude range: 0 to 24.86 N by 0.22 degrees resolution

Author's changes in manuscript: Please see the manuscript at line 105

4. Comments: You make a number of references to differences at the "local" scale - how do these differences impact your results, if at all.

Author's response: Thank you for your comment. The confusion may be due to our English language. We have improved it in this revised version. The PCC gives an overview of the regional coherency of the mean spatial patterns between the model and the observation. However, the PCC needs to be completed with a "local" (or sub-regional) analyses to check where are the main differences and where are the weakest or strongest biases.

5. Comments: Table 1: What is your sample size (how many grid points?)

Author's response: For West Africa, the number of grid points is 17472 (182x96) For Central Sahel, the number of grid points is 2457 (91x27) lon: -9.88 to 9.92 with 0.22-degree resolution lat: 10.12 to 15.84 with 0.22-degree resolution

For Sahel West, the number of grid points is 972 (36x27) lon: -17.8 to -10.1 with 0.22-degree resolution lat: 10.12 to 15.84 with 0.22-degree resolution

For Guinea Coast, the number of grid points =3648 (114x32) lon: -14.94 to 9.92 with 0.22-degree resolution lat: 3.08 to 9.9 with 0.22-degree resolution

Author's changes in manuscript: Please see the manuscript at line 144-146
6. Comments: Fig. 1 and your domains did you only include land points for precipitation and temperature? Author's response: Yes, we computed only over the land grid points.

Author's changes in manuscript: Please see the manuscript at line 161-163

7. Comments: You may want to consider keeping a consistent naming scheme for your experiments - I found it hard to follow at times when you had wet vs. dry years, wet vs. Dry initializations, etc.

Author's response: Thank you for your comment. We agree. What we decided to do in this revised manuscript, is to introduce in the beginning that 2003 and 2004 are respectively wet and dry years and then in the next, we avoid use of "wet or dry years" and focus on wet, dry and control experiments in 2003 and 2004 to refer to soil moisture initial conditions.

Author's changes in manuscript: Please go through the manuscript.

References:

Alexander, L. V., and Coauthors, 2006: Global observed changes in daily climate extremes of temperature and precipitation. J.Geophys. Res., 111, D05109, doi:10.1029/2005JD006290.

Andronova, N. G., and M. E. Schlesinger (2001), Objective estimation of the probability density function for climate sensitivity, J. Geophys. Res., 106(D19), 22,605–22,611.

Anthes, R. A., Kuo, Y. H., Hsie, E. Y., Low-Nam, S., & Bettge, T. W. (1989). Estimation of skill and uncertainty in regional numerical models. Quarterly Journal of the Royal Meteorological Society, 115(488), 763-806.

Dessai, S., X. Lu, and M. Hulme (2005), Limited sensitivity analysis of regional climate change probabilities for the 21st century, J. Geophys. Res., 110, D19108, doi:10.1029/2005JD005919.

Hong S. Y. and Pan H. L.: Impact of soil moisture anomalies on seasonal, summertime
circulation over North America in a regional climate model. J. Geophys. Res., 105 (D24), 29 625–29 634, 2000.

Jaeger, E. B., & Seneviratne, S. I. (2011). Impact of soil moisture–atmosphere coupling on European climate extremes and trends in a regional climate model. Climate Dynamics, 36(9-10), 1919-1939.

Kang, S., Im, E. S., & Ahn, J. B. (2014). The impact of two land-surface schemes on the characteristics of summer precipitation over East Asia from the RegCM4 simulations. International journal of climatology, 34(15), 3986-3997.

Katz, R., and B. Brown, 1992: Extreme events in a changing climate: Variability is more important than averages. Climatic Change, 21, 289–302.

Kiktev, D., D. M. H. Sexton, L. Alexander, and C. K. Folland,2003: Comparison of modeled and observed trends in indices of daily climate extremes. J. Climate, 16, 3560–3571.

Kim J-E., and Hong S-Y.: Impact of Soil Moisture Anomalies on Summer Rainfall over East Asia: A Regional Climate Model Study, Journal of Climate. Vol. 20, 5732–5743, DOI: 10.1175/2006JCLI1358.1, 2006. Knutti, R., T. F. Stocker, F. Joos, and G. K. Plattner (2002), Constraints on radiative forcing and future climate change from observations and climate model ensembles, Nature, 416(6882), 719–723.

Koster, R. D., Mahanama, S. P. P., Yamada, T. J., Balsamo, G., Berg, A. A., Boisserie, M., ... & Guo, Z. (2010). Contribution of land surface initialization to subseasonal forecast skill: First results from a multi-model experiment. Geophysical Research Letters, 37(2). doi:10.1029/2009GL041677.

Rahman, M.M., Lu, M., and Kyi, K. H.: Variability of soil moisture memory for wet and dry basins, J. Hydrol. 523, 107-10118, doi: 10.1016/j.j.jhydrol.2015.01.033, 2015.

Santanello Jr., J. A., P. Lawston, S. Kumar, and E. Dennis, 2019: Understanding the Impacts of Soil Moisture Initial Conditions on NWP in the Context of Land–Atmosphere

**HESSD**

**HESSD**

---

## Author Response (AR1)

**Reply to the comments of editor for the manuscript hess-2020-112**

**Major/Specific Comments:**

*1. Comments:*

I think both reviewers concern regarding choice of only two years for conducting this analysis is valid. I would strongly suggest you to provide a reason for why two years is good enough for these experiments and why you couldn't include more years in the analysis.

*Author's response: Thank you for your comment. We re-run the simulations over 5 years (2001 to 2005) during the months of June to September over our West African domain. We superimposed the 5 years and their climatological average in order to analyze the changes in daily soil moisture over our domain studied (Fig.1). The Fig.1 shows that the weakest and strongest impact of the dry experiments is found for 2003 and 2004 respectively. For a wet year, the impact of drying out soil moisture is quickly erased. While for a dry year the impact of the drying of the soil is accentuated. This meaning that 2003 and 2004 are respectively the wettest and driest years in dry experiment. However, for the wet experiments, the weakest impact is found for 2004, and the strongest impact is found for the years 2001, 2002 and 2004. The wet experiments confirm the result obtained in dry experiments, 2003 and 2004 are wettest and driest years respectively. To conduct our analyzing to estimate the limits of the impact of internal soil moisture forcing on the new dynamical core non-hydrostatic of RegCM4, we have used the two extreme years 2003 and 2004 (resp. the wettest and the driest years) among the 5 years. It is in the same context, several previous studies chosen two extreme years for their sensitivity study of initial soil moisture condition on the models. Hong and al. (2000) use in their study only two years (3 months per year) to investigate the impact of initial soil moisture over the North of America (in the Great Plains) during the two summers, May-June-July (MJJ) 1988 (corresponding to a drought in the Great plains) and MJJ 1993 (correspond to a flooding event). Over Asia, Kim and Hong (2006) in their paper "Impact of Soil Moisture Anomalies on Summer Rainfall over East Asia: A Regional Climate Model Study" used two contrasted years 1997 (below normal precipitation year) and 1998 (above normal precipitation year).*

[Figure]

**Fig.1**: Changes in daily soil moisture for 5 years (2001 to 2005) and their climatological mean during JJAS over West African domain, from dry (ΔDC) and wet (ΔWC) experiments with respect to their corresponding control experiment.

**2. Comments:**

Reviewer 1's comment #5 on the novelty of this study is also another important comment. I'd strongly advise to respond to this comment with sufficient details and rigor.

*Author's response: Thank you for your comment. There are few papers on the impact of soil moisture in Africa, but they were performed with Global Circulation Model (Koster et al. 2004, Douville et al. 2001, Zhang et al. 2008). While it is known that in the region the interactions between land and atmosphere are very important for convection and rainfall, a regional climate model will contribute to better capture local and regional features. The other novelty of this study is that we use the new dynamical core non-hydrostatic of RegCM4. This has never been done before, while 80% of rainfall in the region is associated with mesoscale convective systems. Moreover, it is the first time that the impact of soil moisture initial conditions on mean and extreme of precipitation and temperature in terms of intensity and duration is investigated over West Africa. Finally, this study provides the quantification of the sensitivity of the RegCM4 model to initial soil moisture conditions, which could allow the evaluation and development of the RegCM4 model in the aim to improve sub-seasonal to seasonal forecast skill. The novelty and contribution of this paper has been added in the revised version of the manuscript (conclusion section).*

**3. Comments:**

In a number of cases, in your response to reviewer #2, you are referring to manuscript "Please see the manuscript at.." since at this stage I can't see the revised manuscript, I can't decide if your responses are satisfactory or not. Please include any substantial changes made in the manuscript to your response too.

> *Author's response: Thank you for your comment. Please see the responses to the reviewer#2 below, we included below and where relevant those substantial changes made in the manuscript.*

**Reply to the comments of the referee 2 for the manuscript hess-2020-112**

**1. Comments:**

How were the years 2003 and 2004 chosen? Perhaps show a time series of precipitation anomalies in your study region to highlight why you chose 2003 and 2004. "Dry" and "wet" are very subjective terms. Or consider showing observed precipitation and soil moisture anomalies during 2003 and 2004.

> *Author's response: Thank you for your comment. Our choice of year stems from a long-term study of rainfall anomalies carried out over the sensitive Sahel zone from 1950 to 2004 (cf fig. below), which shows precipitation anomalies of about ± 1.5 mm of rainfall for the consecutive years 2003 (positive anomalies) and 2004 (negative anomalies). The two contrasted years in West Africa (in 2003 wet year and in 2004 dry year compared to the mean 1950-2004 cf fig. below) have been chosen in the aim to assess the results of the sensitivity study, whatever the year studied.*

[Figure]

Choice of soil moisture data:

**2.  Comments:**
Why initialize soil moisture with ERA 20C, which includes only surface forcings of surface pressure and marine winds only? How well does this dataset compare to satellite-derived observations?  ERA 20C isn't really a reanalysis of soil moisture, because the soil moisture doesn't include any observations.  If the authors want an observationally-based soil moisture dataset, they could consider a product like GLEAM (Martens et al., 2017; Miralles et al., 2011).

> *Author's response:  Thank you for your comment. The soil moisture initialization data of the RegCM4 climate model consists of three surface soil moisture datasets: ERA20c, ESACCI and CPC. We performed a sensitivity study of RegCM4 to these three different data sets in simulating the mean and extreme climate of West Africa. The sensitivity test showed good performance in quantitative assessment of temperature and rainfall simulations of ERA20C data over the entire domain of West Africa and its sub-regions. This is what justified our choice for ERA20C.*

**3.  Comments:**
> You show the observations for precipitation, perhaps show observations for soil moisture too. You could show ERA20C (which you added) and maybe GLEAM as an independent dataset.

> *Author's response: Thank you for your comment. The aim of this work is to study the impact of soil moisture initial conditions in simulation of the climate mean and extreme over West Africa. Due to discrepancies between the datasets in West African region, we used two observational data for temperature and precipitation to validate our simulation outputs. We do not seek in this study to validate soil moisture data.*

**4.  Comments from referee 1:**
Using global wilting points and field capacity: In the dry and wet runs the authors use

a uniform wilting point and field capacity everywhere.  But we expect that both the permanent wilting point and the water holding capacity of the soil differ by location (see, for example Figure 6 of Leenaars et al., 2018). These two values will be radically different in the Sahara Desert and southern Nigeria, for example.  The assumptions made here, that wilting point is 0 or that field capacity is 0.489 is likely unrealistic for many locations, and these extreme initial conditions may be affecting the results. The authors need to more completely justify the use of these initial conditions, as opposed to using the maximum and minimum observed values, for example.

> **Author's response:** *Thank you for your comment. We use the minimum and the maximum soil moisture datasets value in our simulation domain. The two values obtained is defined as volumetric fraction ranging from the permanent wilting point ($=0.117*10^{-4}$) to the field capacity ($=0.47$). Previous studies on Asia and North America have been conducted in the same way (Hong and Pan (2000); Kim and Hong (2006)).*

> **Author's changes in manuscript:** *We did this following modification in the manuscript at the Section 2.2 Line 157-161 of the manuscript:*
> *We initialized the dry and wet soil moisture initial conditions (in volumetric fraction $m^3.m^{-3}$) respectively at the wilting point ($=0.117*10^{-4}$) and the field capacity ($=0.489$) derived from ERA20C dataset. The wilting point and the field capacity correspond to the minimum and the maximum values of soil moisture over our simulation domain studied.*

**5. Comments:**

Effect of methods on the analysis: The fact that the wet year (2003) and the dry year (2004) look the same in most graphs when used as the control seems to indicate that they're either quite similar, or that the values for initial soil moisture are incredibly strong, and are overwhelming everything about the dry vs wet year. Are extreme values of soil moisture like this really useful?  If so, the authors need to better justify them.  I understand that this is a sensitivity analysis, but the authors need to contextualize how relevant this sensitivity analysis is to the conditions of the real world.

> **Author's response:** *Thank you for your comment. We recognize that sensitivity experiments such as "wet" and "dry" ones conducted in this study were not intended to simulate real climate since such extremes are very rare. This sensitivity study can provide estimates of the limits of the impact of internal forcing of the soil moisture for the new non-hydrostatic dynamical core of RegCM4.*

> **Author's changes in manuscript:** *We did this following modification in the manuscript at the Section 4 Line 503-507:*
> *We recognize that sensitivity experiments such as "wet" and "dry" ones conducted in this study were not intended to simulate real climate since such extremes are very rare. These kinds of experiments, however, can provide estimates of the limits of the impact of internal forcing of the soil moisture for the new non-hydrostatic dynamical core of RegCM4.*

6. ***Comments:***

Starting at soil moisture of 0 is quite extreme. how, for example the local minimum and maximum soil moisture estimated for the region in the target starting month (June) in an observational dataset as comparison. Showing local min/max for each pixel would demonstrate how the initial conditions used compare to what has historically been experienced.

>***Author's response:*** *Thank you for your comment. Well, that's what we did with our domain of simulation.*

>***Author's changes in manuscript:*** *We did this following modification in the manuscript at Section 2.2 Line 159-161:*
>*The wilting point and the field capacity correspond to the minimum and the maximum values of soil moisture over our studied simulation domains.*

**References:**

Hong S-Y. and Pan H. L.: Impact of soil moisture anomalies on seasonal, summertime circulation over North America in a regional climate model. J. Geophys. Res., 105 (D24), 29 625–29 634, 2000.

Kim J-E., and Hong S-Y.: Impact of Soil Moisture Anomalies on Summer Rainfall over East Asia: A Regional Climate Model Study, Journal of Climate., Vol. 20, 5732–5743, DOI: 10.1175/2006JCLI1358.1, 2006.

---

## Author Response (AR2)

**Reply to Editor on HESS-112 followed by answers to comments of Referee 1 and Referee 2**

**Editor Decision: Publish subject to revisions (further review by editor and referees)** (16 Nov 2020) by Shraddhanand Shukla
Comments to the Author:
Dear Authors,

I have now received the reviews from both original reviewers. Although both reviewers are recommending Accept after minor revisions, in my opinion at least some of their comments (which are valid) are more in moderate category than minor. In particular I want to highlight the following comment from reviewer #2, which is an important one about clarifying the scope of this study.

"In the concluding remarks the authors seem to imply that this experiment was to estimate the limits of the impact of internal forcing and these are more idealized results. This makes more sense to me, but you must note that these are more idealized results. I think it might be best to state that very prominently in the introduction in order to guide the reader through what you're achieving and the limitations of this study."

The manuscript can only be formally accepted for publication after reviewers comments are fully and satisfactorily addressed. Please carefully go through reviewers comments and provide detailed response and revise manuscript accordingly. As per reviewers availability I'd definitely try to seek their review again.

Thanks again and I look forward to seeing the revised version of this manuscript.

Shrad

**Author response to the editor comments**

*Dear Editor*

*Thank you for your comments and to the reviewers who contributed with their comments to improve the manuscript.*

*As suggested, we have sent the manuscript for an English Language Editing (please see the certificate at the end of this document to ease the reading and to avoid confusion due to language issue.*

*Regarding comment of reviewer # 2, We did this following modification in the manuscript at the introduction line 78 to 83: This study aims to estimate the limits of the impact of internal forcing of initial soil moisture over West Africa region using a Regional Climate Model. Experiments carried out are sensitivity studies that give idealized results of the effect of the initial soil moisture. In this study (part I), the sensitivity of mean climate simulation to initial "wet" and "dry" soil moisture conditions is investigated.*

*Thank you again and best wishes*

*Arona*

**CERTIFICATE OF ENGLISH EDITING**

This document certifies that the paper listed below has been edited to ensure that the language is clear and free of errors. The edit was performed by professional editors at Editage, a division of Cactus Communications, in cooperation with Taylor & Francis Group. The intent of the author's message was not altered in any way during the editing process. The quality of the edit has been guaranteed, with the assumption that our suggested changes have been accepted and have not been further altered without the knowledge of our editors.

**Title**

Influence of initial soil moisture conditions in a regional climate model study over West Africa: Part 1: Impact on the climate mean

**Authors**

Brahima Koné, Arona Diedhiou, Adama Diawara, Sandrine Anquetin, N'datchoh Evelyne Touré, Adama Bamba, and Arsene Toka Kobea

**Order No.**

RODIE_1

EDITINGSERVICES
Supporting Taylor & Francis authors

Signature

*Vikas Narang*

Vikas Narang,
Chief Operating Officer,
Editage

Date of Issue
**November 30, 2020**

[Figure]

**Reply to the comments of referee 1 on HESS-112**

The authors have responded to all initial comments and have, for the most part, revised the manuscript accordingly and satisfactorily. I have a few comments below that either still need further input or are new comments. Overall the manuscript still needs minor revision before publication. One of the weaknesses in the paper still remains in its use of the English language, **there are a number of sentences that do not make sense** or are run-ons. Additionally, I think the authors could still **add some description fo their introduction and summary/conclusion** to make the paper flow better.

**Major/Specific Comments:**
**1. Comments:**
One of the main concerns of both reviewers and the editor was your use of only 2 years in the study. **I don't understand your reasoning noted in your response to the editor that since the sensitivity to initial soil moisture anomalies is only one season you only need two experiments.** I still find this concerning, though the authors have at least provided references that show this experiment design in use in other papers.
In the concluding remarks **the authors seem to imply that this experiment was to estimate the limits of the impact of internal forcing and these are more idealized results. This makes more sense to me, but you must note that these are more idealized results. I think it might be best to state that very prominently in the introduction in order to guide the reader through what you're achieving and the limitations of this study.**

It may also be prudent to include the Figure with "wet" and "dry" years that was included in the author responses to give the reader some indication of how "extreme" these years were. This Figure wasn't included in my review report for some reason, so I'm not sure if its the same as something already included.

> **Author's response**: *Thank you for your comment. We indicated in the introduction your suggestion "that these experiments were to estimate the limits of the impact of internal forcing and these are more idealized results". The figure with "wet" and "dry" years were also included in the manuscript as recommended.*

> **Author's changes in manuscript**: *We did this following modification in the manuscript at the introduction line 78 to 83:* This study aims to estimate the limits of the impact of internal forcing of initial soil moisture over West Africa region using a Regional Climate Model. Experiments carried out are sensitivity studies that give idealized results of the effect of the initial soil moisture. In this study (part I), the sensitivity of mean climate simulation to initial "wet" and "dry" soil moisture conditions is investigated. In part II of the article, the influence of initial soil moisture conditions on climate extremes will be explored.

**2. Comments:**
With respect to my initial comment 3 on the normal distribution. I am aware that the use of the normal assumption is not new. My question is whether you think it is appropriate to use that assumption here. For example, precipitation is better approximated with a gamma distribution on seasonal timescales, etc. The approximation of normal can be problematic for variables that are not normal, and this may be the case here.

*Author's response*: *Thank for your comments. There is a little literature on the probability distributions of annual, seasonal and monthly precipitation in the study region. Markovic (1965) investigated the probability distribution of annual precipitation in the western USA and southwestern Canada using the chi-squared statistic to measure the goodness of fit of sample data to selected probability distribution. He concluded that annual precipitation can be best approximated by the 2-parameter lognormal (LN2) and gamma (GAM) distributions. Sheng Yue and Michio Hashino in their work over Japon suggested that the Pearson type III (P3) and the log-Pearson type III (LP3) distributions are acceptable distribution types to represent statistics of precipitation in Japan with the LN3 distribution as a potential alternative. These studies make an attempt to determine the probability distribution types of annual, seasonal and monthly precipitation across these regions. However, such studies have never been done over West Africa. You are right, the GAM distribution was frequently used to represent monthly and seasonal precipitation (Ropelewski et al.,1985; Wilks & Eggleston, 1992), but it is worth to note that the Gamma distribution* **is useful for variable which is always positive** *(Than et al. (2017), Jaeger and* Jaeger and *Seneviratne, 2011). However, for the biases or changes (including positive and negative values), a normal mode type distribution is more suitable (Gao et al., 2016).*

**3. Comments:**

You have an entire section on how your experiments influence surface fluxes but only devote a line to these results in the concluding remarks, which makes me think that they are less important. Please add some context to the concluding remarks on why these results are significant to this experiment design and the field.

*Author's response:* *Thank you for this comment which helps to improve the reading of the manuscript and the presentation of the results.* We added a paragraph in the concluding remarks on these significant results in the context of our experiments.

**Author's changes in manuscript**: *We did this following modification in the manuscript at the conclusion line 473 to 478:* Our study showed significant impact of initial soil moisture conditions anomalies on the surface energy fluxes. We observed in wet (dry) experiments that the cooling (warming) of surface temperature was associated with an increase (decrease) of sensible heat flux, a decrease (increase) of latent heat and an increase (decrease) of the depth of the boundary layer over the region, with different magnitudes varying from one sub-region to another.

**4. Comments:**

Similarly I think your end of section summaries could be improved and put into the context of your experiments. For example, line 447 - 450 you discuss the cooling and warming of surface temperature, but what does that have to do with your wet, dry, etc. experiments?

*Author's response:* *Thank for your comment. You are right. We rewrote the section summaries to put them into the context of our experiments.*

***Author's changes in manuscript:*** *We did this following modification in the manuscript at the section 3.2 line 435 to 440:* Summarizing the results of this section, in the wet experiments, the cooling of mean surface temperature is associated with a decrease of latent heat flux, an increase in sensible heat flux and the PBL depth over most of the studied domain. Conversely, in the dry experiments, the warming of surface temperature is associated with an increase of the latent heat flux, a decrease of the sensible heat flux and PBL height.

*Minor Comments:*

1. ***Comment:*** *Line 41-45: References are needed.*

   ***Author's response:*** *Thank for your comment. We added reference.*

   ***Author's changes in manuscript:*** *We did this following modification in the manuscript at the section 1 line 45 to 49:* Schär et al. (1999) sustained that the role of soils may be comparable to that of the oceans. The solar energy received by the oceans is stored in summer and used to heat the atmosphere in winter. The precipitation received by the soils is stored in winter and contributed to moisten and cool the atmosphere in summer.

2. ***Comment:*** *Avoid language like "tends to cause", etc. as it lessens the impact of the results.*

   ***Author's response:*** *Thank for your comment. As suggested, we have sent the manuscript for an English Language Editing (please see the certificate at the end of this document) and these expressions have been removed and changed in this revised version.*

   ***Author's changes in manuscript:*** *Please see the revised manuscript.*

3. ***Comment:*** *You've introduced some abbreviations and didn't use them continuously, please make sure you did this. E.g. page 7 "mean bias" vs. "MB".*

   ***Author's response:*** *Thank very much for this remark. We used the abbreviation introduced in the whole revised manuscript.*

   ***Author's changes in manuscript:*** *Please see the revised manuscript.*

4. ***Comment:*** *Line 230: What do you mean by "peak mode of change"? You had been previously saying "peak of change" - Which I'm also not sure what that means. Do you mean "maximum change"?*

   ***Author's response:*** *Thank for your comment. Yes, we mean by "peak mode of change" the "maximum magnitude of change". We have sent the manuscript for revision in English and these are the expressions that have been reported.*

   ***Author's changes in manuscript:*** *We replace "peak of change" by maximum change to make it more comprehensive. Please see through the revised manuscript.*

5. ***Comment:*** *Grammatical Comments: I'm not going to list all of these, but I suggest you still revise your manuscript for English language. There are a number of places that the sentences don't make sense, are "clunky", or just are incomplete. A few examples are below:*

*Author's response: Thank for your comment. We revised the manuscript for English language and take in account all your remarks. Please see the Certificate confirming that the issue with English Language has been addressed.*

*Line 36: "The strength of soil moisture impact on land-atmposphere coupling is variable according to the place and with the season." You can simply say: "The strength of soil moisture impacts on land-atmosphere coupling varies according to location and season" and this will greatly improve the clarity of the sentence.*

*Author's response: Thank you very much for your suggestion. Please check lines 42-43*

*Line 91: "The scheme of the large-scale precipitation used is from Pal et al. (2000), the moisture scheme is the SUBEX (SUBgrid EXplicit moisture scheme) takes in account the cloud variability scale sub-grid, and the accretion processes and evaporation for stable precipitation following the work of Sundqvist et al., 1989." Can be restated: "The large-scale precipitation scheme is from Pal et al. (2000) and the moisture scheme is the SUBEX (SUBgrid EXplicit moisture scheme). The SUBEX take into account the sub-grid scale cloud variability, and the accretion processes and evaporation for stable precipitation following the work of Sundqvist et al., 1989."*

*Author's response: Thank you for your proposition. Done. Please check lines 99-102*

*Line 114: The uncertainties reduction related to the absence of reliable observation system over the region (Sylla et al., 2013a; Nikulin et al.,2012), we validated the simulated precipitation based on two products..." I'm not sure what this means.*

*Author's response: Thank you. The sentence was reworded as follows:* Due to the coarse resolution of the climate observing network over the region, we validated the simulated precipitation based on two satellite derived products (Sylla et al., 2013a; Nikulin et al.,2012): *Please check lines 122-124.*

*Line 172: "In the aim to identify the extreme years (driest and wettest) impacted by the dry and wet experiments among the five years simulations (2001 to 2005), we display Changes in daily soil moisture for 5 years (2001 to 2005) and their climatological mean during JJAS over West African domain, from dry and wet experiments with respect to their corresponding control experiment in Figure 2." This is a run-on and clarity would be increased if it were simplified.*

*Author's response: Thank you. We reworded as follows. Please check lines 178-183:* To identify the extreme years (driest and wettest) impacted by the dry and wet experiments among the simulation period (2001–2005), we determined changes in daily soil moisture and their climatological mean during JJAS over the West African domain from dry and wet experiments with respect to their

corresponding control experiment. These changes are presented in Fig. 2, which shows that the weakest and strongest impacts of the dry experiments were observed in 2003 and 2004, respectively.

*Line 188: "...for JJAS 2003 and JJAS 2004 and their corresponding simulated from control experiments..." And their corresponding what?*

> **Author's response:** *Thank you.* "…we determined changes in daily soil moisture and their climatological mean during JJAS over the West African domain from dry and wet experiments with respect to their corresponding control experiment." Please check line 181.

**References:**

*Gao, X.-J., Shi, Y., and Giorgi, F.: Comparison of convective parameterizations in RegCM4 experiments over China with CLM as the land surface model, Atmos. Ocean. Sci. Lett., 9, 246–254, https://doi.org/10.1080/16742834.2016.1172938, 2016.*

*Jaeger E. B., and Seneviratne S. I. : Impact of soil moisture-atmosphere coupling on European climate extremes and trends in a regional climate model, Clim. Dyn., 36(9-10), 1919-1939, doi:10.1007/s00382-010-0780-8, 2011.*

*Markovic, R. D. (1965) Probability of best fit to distributions of annual precipitation and runoff. Hydro. Paper no. 8, Colorado State Univ., Fort Collins, Colorado, USA.*

*Ropelewski, C. F., Janowiak, J. E. & Halpert, M. S. (1985) The analysis and display of real time surface climate data. Monthly Weather Review 113, 1101–1106.*

*SHENG YUE & MICHIO HASHINO (2007) Probability distribution of annual, seasonal and monthly precipitation in Japan, Hydrological Sciences Journal, 52:5, 863-877, DOI: 10.1623/hysj.52.5.863*

*Thanh N.-D., Fredolin T. T., Jerasorn S., Faye C., Long T.-T., Thanh N.-X., Tan P.-V., Liew J., Gemma N., Patama S., Dodo G. and Edvin A.: Performance evaluation of RegCM4 in simulating extreme rainfall and temperature indices over the CORDEX-Southeast Asia region. Int. J. Climatol. 37: 1634–1647. Published online 28 June 2016 in Wiley Online Library (wileyonlinelibrary.com) DOI: 10.1002/joc.4803, 2017.*

*Wilks, D. S. & Eggleston, K. L. (1992) Estimating monthly and seasonal precipitation distributions using the 30- and 90-day outlooks. J. Climate 5, 252–259.*

*Schär, C., Lüthi, D., Beyerle, U. & Heise, E. The soil-precipitation feedback: A process study with a regional climate model. J. Clim. 12, 722–741 (1999).*

**CERTIFICATE OF ENGLISH EDITING**

This document certifies that the paper listed below has been edited to ensure that the language is clear and free of errors. The edit was performed by professional editors at Editage, a division of Cactus Communications, in cooperation with Taylor & Francis Group. The intent of the author's message was not altered in any way during the editing process. The quality of the edit has been guaranteed, with the assumption that our suggested changes have been accepted and have not been further altered without the knowledge of our editors.

**Title**

Influence of initial soil moisture conditions in a regional climate model study over West Africa: Part 1: Impact on the climate mean

**Authors**

Brahima Koné, Arona Diedhiou, Adama Diawara, Sandrine Anquetin, N'datchoh Evelyne Touré, Adama Bamba, and Arsene Toka Kobea

**Order No.**

RODIE_1

**EDITINGSERVICES**
Supporting Taylor & Francis authors

Signature

*Vikas Narang*

Vikas Narang,
Chief Operating Officer,
Editage

Date of Issue
**November 30, 2020**

[Figure]

**Taylor & Francis Editing Services**

www.tandfeditingservices.com
support@tandfeditingservices.com

**Reply to the comments of referee 2 on HESS-112**

I appreciate the edits the authors have made to the manuscript. The authors, however, did not address one of my main concerns, which was that even as a sensitivity analysis their choice of permanent wilting point and field capacity may be rather extreme. I believe either the authors misunderstood my comment, or I am misunderstanding their methods (either is possible).

In my comment 4 I expressed concern about the authors using a global value for wilting point and field capacity, because in reality both of these values change dramatically within their domain. The authors cite the methods of Hong and Pan (2000), but in Section 2.2. of that study the authors say that "the version of soil model used in this study uses a uniform vegetation fraction and soil texture value over land", which would produce global values of field capacity and wilting point in their model, making the use of single global values (0.1 and 0.47) analogous to the use of local values.

If the model that the authors use in this current study does not use a uniform soil texture and vegetation fraction, then the soil model would not have uniform values for wilting point and field capacity. I am not familiar with the model so I cannot say if this is the case. I would ask that the authors clarify for the reader either that (1) their soil model uses uniform values of wilting point and field capacity (as does Hong and Pan, 2000) or (2) be clear that even though their soil model has spatially varying values of field capacity and wilting point, that they use a single global value. I ask this only so that readers will be able to follow the methods the authors are using.

Lines 18-20 in the abstract stating that "we initialized the soil moisture at the wilting points and field capacity with dry and wet soil moisture initial conditions", for example, is untrue if the authors are using a model that has spatially varying wilting points and field capacities, but initializing all locations using a single value. The authors could instead say that they initialize the soils at volumetric fractions of 0 and 0.49 everywhere, which would be more clear for the reader.

> *Author's response:* *Thank you very much for your comment. You are right. As confusion may come from the issue with English language, the manuscript has been deeply edited to ease the reading. The line has been reworded in the abstract.*

> *Author's response:* *Thank for your suggestion. We did this following modification in the manuscript at line 161-163:* We initialized the dry and wet soil moisture initial conditions (in volumetric fraction $m^3.m^{-3}$) respectively at the minimum value (=$0.117*10^{-4}$) and the maximum value (=0.489) derived from ERA20C dataset over the West Africa studied domain.

**CERTIFICATE OF ENGLISH EDITING**

This document certifies that the paper listed below has been edited to ensure that the language is clear and free of errors. The edit was performed by professional editors at Editage, a division of Cactus Communications, in cooperation with Taylor & Francis Group. The intent of the author's message was not altered in any way during the editing process. The quality of the edit has been guaranteed, with the assumption that our suggested changes have been accepted and have not been further altered without the knowledge of our editors.

**Title**

Influence of initial soil moisture conditions in a regional climate model study over West Africa: Part 1: Impact on the climate mean

**Authors**

Brahima Koné, Arona Diedhiou, Adama Diawara, Sandrine Anquetin, N'datchoh Evelyne Touré, Adama Bamba, and Arsene Toka Kobea

**Order No.**

RODIE_1

EDITINGSERVICES
Supporting Taylor & Francis authors

[Figure]

Signature

*Vikas Narang*

Vikas Narang,
Chief Operating Officer,
Editage

Date of Issue
**November 30, 2020**

**Taylor & Francis Editing Services**

www.tandfeditingservices.com
support@tandfeditingservices.com

---

## Author Response (AR3)

**Reply to the comments of referee 1 for Second Revision of HESS-112**

Thank you for revising the manuscript. I am still recommending major revisions due to the important comments raised by reviewer #2. Please make sure to carefully reviewer #2's comment and clarify/revise accordingly. Of particular significance is their comment on significance test. Please address it carefully and clearly explain how you are conducting the significance test. The reviewer also found several errors (grammatical etc), so please carefully review for those too.

**Major/Specific Comments:**

1. **Comments:**

   I appreciate that the authors took the time to send the manuscript for English language editing, but I found a mistake in the first sentence of the paper, and a number of oddities in the first paragraph. While the authors sent this out for editing they should still re-read it for flow and inconsistencies.

   *Thank you for your comments. We proofread the manuscript for fluidity and corrected any inconsistencies.*

   (a) In the climate system, soil moisture is a crucial variable that influence[s] water balance and surface energy components...

   *Line 37: Thank you. Done*

   (b) Line 47: "Another potentially useful earth system component that varies slowly is soil moisture" - You've already introduced soil moisture. It seems like you're introducing it again. You actually could just get rid of this sentence.

   *Line 47: Thank you for your suggestion. We have deleted this sentence as advised.*

   (c) "One important role of anomalies in soil moisture in the coupling of land and atmosphere has been shown in several studies, using numerical climate models (Jaeger and Seneviratne, 2011; Zhang et al., 2011) and observation datasets (Zhang et al., 2008a; Dirmeyer et al., 2006)." What is the important role?

   *Line 53-59 : Thank you. In this revised version, we removed the reference to Jaeger and Seneviratne (2011) which referred on climate extremes (already cited in Part 2) and we added a sentence to illustrate the important role of soil moisture as follows: **The important role of anomalies in soil moisture in the coupling between land and atmosphere has been shown in several studies, using numerical climate models (Zhang et al., 2011) and observation datasets (Zhang et al., 2008a; Dirmeyer et al., 2006). For instance, over East Asia, Zhang et al., (2011) showed that soil moisture is found to have a much stronger impact on daily maximum temperature variability than on daily mean temperature variability, but generally has small effects on daily minimum temperature, except in the***

*eastern Tibetan Plateau. They showed that soil moisture has a prominent contribution to precipitation variability in many parts of western China.*

(d) However, at local and regional scales, the land-atmosphere coupling studies with AGCM[s]...

*Thank you, done. Please check lines 66-68; **However, at local and regional scales, the land-atmosphere coupling studies with AGCMs, present significant uncertainties (Xue et al. 2010).***

(e) Line 65, define RCMs.

*Thank you, done. Please check lines 67-68 : **The regional climate models (RCMs) have been used to simulate the impact on interannual climate variability of anomalies in soil moisture…***

f) Overall, the results of these studies show that, during summer, the strong impact of the anomalies of soil moisture in land-atmosphere occurred mainly over the transition zones with a climate between wet and dry climate regimes, [in agreement with Koster et al.]

*Thank you for the suggestion. Done lines 73-76: **Overall, the results of these studies showed that, during summer, the strong impact of the anomalies of soil moisture in land-atmosphere occurred mainly over the transition zones with a climate between wet and dry regimes, in agreement with Koster et al. (2004).***

(g) Line 75: This study will focus on the influence of initial soil moisture conditions anomalies. Remove either conditions or anomalies.

*Thank you very much. The sentence has been reword as follows (line 78): **This study will focus on the influence of soil moisture initial conditions on climate mean.***

... Again, I'm not listing all of these instances. The authors need to be more proactive in their read-through of the paper.
*Yes, we took the time to go through the document, correcting typos and making it easier to read.*

**2. Comments:**

Perhaps I missed it but I'm still confused about the experiment design. Are these ensemble experiments? I.e. does your wet initial soil moisture condition experiment (for example) have ensemble members of precipitation and temperature, or is it just one model run? I just want to make sure I'm understanding this correctly. I might note somewhere that you have 1 realization of precipitation and temperature (or its an ensemble) for each soil moisture initial condition just to make this abundantly clear.

*Thank for your comments. In this revised version, we began the section 2.2 (Experiments setup and analysis methodology) as follows (lines 130-145):* **The European 20th Century Weather Prediction Center ERA20C soil moisture reanalysis was used to initialize the control experiment, while its domain-wide minimum and maximum values were used to establish the initial dry and wet soil moisture conditions respectively (hereafter dry and wet experiments). We initialized the dry and wet soil moisture initial conditions (in volumetric fraction m3.m-3) respectively at the minimum value (=0.117*10-4) and the maximum value (=0.489).**

**We designed three experiments (reference, wet, and dry), each with an ensemble of five (5) simulations starting from June 1st to September 30th. The difference between these three experiments is the change in the initial soil moisture condition (reference initial soil moisture condition, wet initial soil moisture condition, and dry initial soil moisture condition) during the first day of the simulation (June 1st 2001, 2002, 2003, 2004 and 2005) over the West African domain. Then, we selected the two runs most impacted by the wet and dry soil moisture initial conditions in order to exhibit the effects on the climate mean beyond the limits of the impacts of RegCM4 initial soil moisture internal forcing. In the same context, several previous studies have selected two extreme years to investigate the climate models sensitivity to soil moisture initial conditions (Hong et al., 2000; Kim and Hong, 2006) outside Africa.**

3. **Comments:**

Are the soil moisture initial conditions soil moisture anomalies? Make sure this language is consistent.

*Thank you. We used in the whole manuscript (wet or dry) soil moisture initial conditions. See for example at lines 159, 161, 166,...*

4. **Comments:** This comment goes for both hess112 and hess113. I'm not entirely satisfied by your statistical significance test using a t-test. If I'm understanding your experiment design and statistical significance test correctly, this is a two-sample t-test of means - You've compared your control (sample 1) to the sensitivity test (sample 2). Are you doing this for 1 year only, for your entire study period? For a two sample t-test you need to assume the values are independent, and I'm wondering if they truly are independent samples. Both reviewers have made notes about your significance test asking specifically for the sample size used and it has not been adequately provided. Some of these questions might be answered if the authors better explained their significance test procedure.

*Thank you very much. The main concern was that we performed the significance test with monthly values leading to samples of small size. We agree that this is a shortcoming.*
*In this revised version, instead of doing the Student t-test with monthly means, we did it with daily values (from June to September) for each year (2003 and 2004) and thus, with samples of 115 days (without the 7 days spin-up period).*

*For each year, the Student t-test is used to compare the significance of the difference between a wet or dry sensitivity test (sample 1) and the control (sample 2) in assuming that our two samples are independent and in considering that this method performs well for climate simulations compared to more sophisticated techniques developed to address autocorrelation (Damien et al., 2014).*

Damien Decremer, Chul E. Chung, Annica M. L. Ekman & Jenny Brandefelt (2014) Which significance test performs the best in climate simulations?, **Tellus A: Dynamic Meteorology and Oceanography**, 66:1, DOI: 10.3402/tellusa.v66.23139

*The t-test is extensively used for analysis in climate sciences; it is fairly robust and easy to use and interpret (Wu et al., 2020; Menedez et al., 2019; Talahashi and Polcher, 2019; Wu et al., 2019; Alvdes et al., 2017):*

Menéndez, C. G., Giles, J., Ruscica, R., Zaninelli, P., Coronato, T., Falco, M., ... & Li, L. (2019). Temperature variability and soil–atmosphere interaction in South America simulated by two regional climate models. **Climate Dynamics**, *53*(5), 2919-2930.

Takahashi, H. G., & Polcher, J. (2019). Weakening of rainfall intensity on wet soils over the wet Asian monsoon region using a high-resolution regional climate model. **Progress in Earth and Planetary Science**, *6*(1), 1-18.

Wu, T., Hu, A., Gao, F., Zhang, J., & Meehl, G. A. (2019). New insights into natural variability and anthropogenic forcing of global/regional climate evolution. **Nature, Npj Clim. Atmos. Science**, *2*(1), 1-13. https://doi.org/10.1038/s41612-019-0075-7

Alves, L. M., Marengo, J. A., Fu, R., & Bombardi, R. J. (2017). Sensitivity of Amazon regional climate to deforestation. **American Journal of Climate Change**, *6*(1), 75-98.

Wu, J., Han, Z., Xu, Y., Zhou, B., & Gao, X. (2020). Changes in extreme climate events in China under 1.5 C–4 C global warming targets: Projections using an ensemble of regional climate model simulations. **Journal of Geophysical Research: Atmospheres**, *125*(2), e2019JD031057.

*The t-test takes into account, the difference between the means of each sample, the variance (S) and the number of degrees of freedom (n − 1), which depends on the sample size (n). The test statistic is calculated as:*

$$t = \frac{\bar{X}_1 - \bar{X}_2}{\sqrt{\dfrac{S_1^2}{n_1} + \dfrac{S_2^2}{n_2}}}$$

*Where $\bar{X}_1$ ($\bar{X}_2$) are the sample means, $n_1$ ($n_2$) are the sample sizes and $S_1^2$ ($S_2^2$) are the sample variances. The t-test at the 95% confidence level was used to consider statistically significant.*

*Author's changes in the manuscript: In this revised, we added 2 paragraphs on the Student t-test used in this study as follows (lines 166-179):*

*For the two years most sensitive to soil moisture initial conditions, the Student t-test is used to compare the significance of the difference between a wet or dry sensitivity test (sample 1) and the control (sample 2) in assuming that our two samples are independent and in considering that this method performs well for climate simulations compared to more sophisticated techniques developed to address autocorrelation (Damien et al., 2014). The Student t-test is extensively used for analysis in climate sciences; it is fairly robust and easy to use and interpret (Menedez et al., 2019; Talahashi and Polcher, 2019). The Student t-test takes into account, the difference between the means of each sample, the variance (S) and the number of degrees of freedom (n − 1), which depends on the sample size (n). The test statistic is calculated as:*

$$t = \frac{\bar{X}_1 - \bar{X}_2}{\sqrt{\dfrac{S_1^2}{n_1} + \dfrac{S_2^2}{n_2}}}$$

*Where $\bar{X}_1$ ( $\bar{X}_2$) are the sample means, $n_1$ ( $n_2$) are the sample sizes and $S_1^2$( $S_2^2$) are the sample variances. In this study, the t-test at the 95% confidence level was used to consider statistically significant.*

5. **Comments:** Line 440: The strongest precipitation decrease (increase) is found over the central Sahel (over central Sahel) in dry (wet) experiment in JJAS 2003 (JJAS 2004) with maximum change reaching -4% (40%). You don't need that parenthetical reference for the central Sahel twice. In addition, and I think I've noted this in some of my earlier reviews, over-use of this style of writing can be very confusing for long sentences. You could just make this into two sentences instead of using the () style.

*Thank for your comment. You are right. We rewrote and corrected the sentence to make it more comprehensive. For long sentences, we make them into two sentences instead of using the () style.*

*In this revised version the sentences have been changed as follows:*

*Lines 298-302: **The strongest precipitation decrease was found over west Sahel for dry experiment for JJAS 2003 with maximum change reaching −4%. While, the strongest precipitation increase was found over the central Sahel for wet experiment in JJAS 2004 with maximum change about 40%.***

*We improved also the lines 360-363 which were with the () style: **In the wet experiments, the strongest latent heat flux increase is found over West Sahel with maximum change reaching 36.49 W.m-2 in JJAS 2004 (Table2). In the dry experiments, the strongest latent heat flux decrease is located over Guinea Coast with maximum change reaching −14.64 W.m-2 in JJAS 2004 (Table2).***

*Lines 425-429: **In the dry experiment, the strongest precipitation decrease is found over the Central Sahel in JJAS 2003 with maximum change reaching −4% while in the wet experiment,***

*the strongest precipitation increase is found over the West Sahel in JJAS 2004 with maximum change reaching 40%.*

6. **Comments:** It would be good to re-iterate how these results fit into the greater body of literature in your concluding remarks. For example, how this has not been quantified for West Africa, and these are very idealized experiments meant to provide what is essentially a first look or guess of what the impacts of soil moisture initial conditions might be in the region. These results are also likely very specific to RegCM4. The authors could also speculate about further work that could be done given the results of these experiments

*Thank for the suggestion. We have improved the conclusions in adding a paragraph on the perspectives as follows (452-460):*

**This study is the first investigating the impact of soil moisture initial conditions in West Africa. However, this study is based on idealized experiments and very specific to RegCM4. In the future, an investigation using different RCMs in a multi-model framework will contribute to better quantify the impact of soil moisture initial conditions. At shorter timescales, there is a need to understand how the soil moisture initial conditions contribute to the triggering and the maintenance of the mesoscale convective systems which are known to explain large amount of rainfall in the region (Mathon et al., 2002).**

Mathon, V., Diedhiou, A., & Laurent, H. (2002). Relationship between easterly waves and mesoscale convective systems over the Sahel. *Geophysical research letters*, *29*(8), 57-1.

---

## Author Response (AR4)

**Reply to the comments of referee 1 for Second Revision of HESS-112**

I found the manuscript much improved from the previous iteration, particularly with respect to readability, and the authors have addressed my comments satisfactorily. My final comments are minor suggestions that would further strengthen the text. I recommend minor revisions at this stage.

**Minor Comments:**

**1. Comments:**

The authors did a great job revising the manuscript for editorial issues. I found some lingering errors, and I've noted a glaring errors here, but I'm hoping the journal editor can solve the remainder.

Line 233-234: The strongest precipitation increase is found over West Sahel for the wet experience... Should read "experiment".

*Author's response*: Thank you for your comment. Yes, you're right. We checked both manuscripts and changed "experience" to "experiment".

**Author's changes in manuscript:**

Line 235: The strongest precipitation increase is found over West Sahel for the wet experiment. Line 337: The strongest mean temperature decrease is observed over the Central and West Sahel in wet experiments with the maximum change approximately -1.5 °C (Table 2).

**2. Comments:**

In Fig 5, Fig 13, 15, 17, 19, 21 you make a reference to panels, but the panels are not labeled.

Author's response: Thank you. Done now.

Author's changes in manuscript: Please see at the Fig 5, 13, 15, 17, 19 and 21.

**3. Comments:**

It may be worthwhile to add a brief sentence on how your ensemble members are generated.

*Author's response:* Thank you for your suggestion. In this revised version, we added 3 sentences in section 2.2 at lines 138 to 143.

**Author's changes in manuscript:**

Lines 138 to 143: We designed three experiments (reference, wet, and dry), each with an ensemble of five (5) simulations. The simulation time period for each experiment lasts for 4 months, starting from June 1st to September 30th. The difference between these three experiments is the change in the initial soil moisture condition (reference initial soil moisture condition, wet initial soil moisture condition, and dry initial soil moisture condition) during the first day of the simulation (June 1st 2001, 2002, 2003, 2004 and 2005) over the West African domain.

**4. Comments:**

I think you may be missing an interesting conclusion stemming from this work (though maybe

you've alluded to this and I missed it). In e.g. fig 5, the regional PDF shifts for dry and wet soil moisture are in the central Sahel are quite striking (what should be panel a) when compared to the West Africa region as a whole. Many of the other supporting Figures show regional PDF differences that are reflected in West Africa. I'm just curious if you have more to say on this.

Author's response: Thank for your comment. You are right.

Author's changes in manuscript: We added the sentences below in the manuscript at lines 412 to 422: This study shows that when averaged over the entire West African region, the sensitivity of rainfall to initial soil moisture conditions is not captured. However, it is important to have a good initialization of soil moisture because depending on the region, the sensitivity of rainfall can be more or less strong. Indeed, rainfall is more sensitive to initial soil moisture conditions in the western and central Sahel (arid zones) than in the Guinean Coast (humid zone). In these arid Sahelian zones, wetter initial conditions will result in more rainfall, especially in the West Sahel, and dry initial conditions will result in less rainfall, especially in the Central Sahel. In the Guinean Coast, the sensitivity of precipitation to initial soil moisture conditions is lower and other factors could be involved such as moisture advection from the Atlantic by the monsoon flow (Kone et al., 2010) and a lower albedo (Charney 1975).